# Functional and multiscale 3D structural investigation of brain tissue through correlative in vivo physiology, synchrotron microtomography and volume electron microscopy

Carles Bosch [1✉], Tobias Ackels [1,2], Alexandra Pacureanu [1,2,3], Yuxin Zhang [1,2], Christopher J. Peddie [4], Manuel Berning[5,6], Norman Rzepka [6], Marie-Christine Zdora[7,8,9], Isabell Whiteley [1,2], Malte Storm [8,10], Anne Bonnin [11], Christoph Rau[8], Troy Margrie[12], Lucy Collinson [4] & Andreas T. Schaefer [1,2✉]

Understanding the function of biological tissues requires a coordinated study of physiology and structure, exploring volumes that contain complete functional units at a detail that resolves the relevant features. Here, we introduce an approach to address this challenge: Mouse brain tissue sections containing a region where function was recorded using in vivo 2-photon calcium imaging were stained, dehydrated, resin-embedded and imaged with synchrotron X-ray computed tomography with propagation-based phase contrast (SXRT). SXRT provided context at subcellular detail, and could be followed by targeted acquisition of multiple volumes using serial block-face electron microscopy (SBEM). In the olfactory bulb, combining SXRT and SBEM enabled disambiguation of in vivo-assigned regions of interest. In the hippocampus, we found that superficial pyramidal neurons in CA1a displayed a larger density of spine apparati than deeper ones. Altogether, this approach can enable a functional and structural investigation of subcellular features in the context of cells and tissues.

[1] Sensory Circuits and Neurotechnology Lab., The Francis Crick Institute, London, UK. [2] Department of Neuroscience, Physiology and Pharmacology, University College, London, UK. [3] ESRF, The European Synchrotron, Grenoble, France. [4] Electron Microscopy STP, The Francis Crick Institute, London, UK. [5] Department of Connectomics, Max Planck Institute for Brain Research, Frankfurt am Main, Germany. [6] Scalable minds GmbH, Potsdam, Germany. [7] Department of Physics and Astronomy, University College London, London, UK. [8] Diamond Light Source, Harwell Science and Innovation Campus, Didcot, UK. [9] School of Physics and Astronomy, University of Southampton, Highfield Campus, Southampton, UK. [10] Institute of Materials Physics, Helmholtz-Zentrum Hereon, Geesthacht, Germany. [11] Paul Scherrer Institut, Villigen, Switzerland. [12] Sainsbury Wellcome Centre, University College London, London, UK. ✉email: carles.bosch@crick.ac.uk; andreas.schaefer@crick.ac.uk

In multicellular organisms, cell ensembles coordinate and specialise into higher-order functional units: tissues, organs, and organisms. Decoding how these units operate requires approaches able to provide both functional and structural insight[1]. Those approaches need to be robust and reliable, and therefore capable of harnessing cohorts large enough to assess different conditions and control for variability. In the brain, this challenge is exemplified by the question of how neuronal circuits compute and transform information[2]. In this case, the volumes to explore should ideally contain a full neuronal circuit[3,4] which in mammals typically reaches the ~mm³ scale. Moreover, the spatial resolution must be sufficient to discriminate synaptic contacts[5,6], ideally requiring (10 nm)³ granularity. Finally, in vivo functional analysis needs to be able to explore different conditions with temporal resolution sufficient to resolve neuronal activity[7]. While in principle such a dataset (with replicates and different conditions) would enable a mechanistic understanding of how a specific neuronal circuit transforms information, there is no single technology that can provide all of that temporal and spatial insight. Conversely, correlative multimodal imaging (CMI) pipelines[8–20] are needed to integrate information from different sources on the same specimen. These approaches simplify data acquisition and management burdens by e.g. enabling targeted high-resolution recordings. They can give orthogonal information of functional/temporal (e.g. functional imaging prior to EM[8]) or molecular[21] nature. Therefore, by designing CMI pipelines the challenge becomes more tractable, namely to find individual technologies capable of providing parts of the required insights while preserving the specimen between applying the different imaging technologies.

Functional information can be reliably obtained from mm³ tissue volumes using fluorescent light microscopy such as in vivo multiphoton imaging techniques[22]. While these techniques do not provide dense structural information sufficient to resolve neuronal connectivity, they can preserve the tissue for possible subsequent correlative high-resolution analysis. For many decades, electron microscopy (EM) has been the dominant technique giving access to such ultrastructural information in cells and tissues[5]. Extending its operation to the 3rd dimension through volume EM (vEM) can in principle deliver the structural insight required to extract computational mechanisms from volumes of tissue[23–33]. It remains, however, highly time consuming and challenging for volumes exceeding ~0.005 mm³ (~months to years of data acquisition and processing times). Moreover, due to the limited penetration depth of electrons in tissue, vEM involves physical sectioning steps and is thus destructive and error-prone. All this together makes contemporary vEM approaches well-suited to obtain individual high-quality datasets for up to (~100–1000 μm)³ volumes[6,34–38]; but they lack the robustness and scalability to routinely combine directly with functional imaging on their own or provide statistical information on the tissue scale from biological replicates.

X-ray computed tomography (CT), i.e. 3D volumetric X-ray imaging, is a technique that can be useful in providing contextual information over large volumes: A single conventional X-ray CT scan can cover larger volumes than most vEM approaches[11]. Furthermore, X-ray imaging relies on contrast mechanisms that are often analogous to those relevant in EM (elements that provide strong back-scattering or absorption signals in EM do also result in strong absorption and phase contrast in X-ray images). This in turn makes sample preparation compatible with both techniques and simplifies correlating datasets obtained by these two modalities. While the spatial resolution of conventional laboratory X-ray μCTs (LXRT) can reach micrometre length scales[39], the broad polychromatic spectra and low spatial coherence of most commercial laboratory systems significantly limit the achievable image contrast and density resolution. This reduces the ability to reliably segment micrometre-sized structures and is only rarely sufficient to identify individual cell bodies[11,39]. In contrast, synchrotron X-ray sources provide the ability to rapidly measure large volumes of a sample or large numbers of samples with good statistics[40]. Moreover, the use of X-ray phase-contrast (PC) imaging methods such as free-space propagation PC[41], grating interferometry[42], or speckle-based PC[43] allows for measuring unstained biomedical samples in near-native state. Staining can be applied to further improve contrast[44–47], and coherent imaging techniques such as X-ray holography[48] or ptychography[49–51] can bring the spatial resolution for biological tissue to the sub-100 nm level.

Here we demonstrate that synchrotron X-ray computed tomography with propagation-based phase contrast (SXRT) can be used to obtain histological information from samples prepared for subsequent EM over length scales of several millimetres. The partially coherent X-rays allow for the use of phase-contrast imaging approaches[52,53] enabling a spatial resolution sufficient to densely resolve neurites. The use of hard X-rays (with an energy higher than approximately 10 keV) in turn makes it possible to use heavy-metal-rich stains that also provide sufficient contrast for subsequent SBEM imaging. Finally, SXRT, which is capable of investigating samples as large as several mm³ at subcellular spatial resolution, combined with LXRT[11], which provides large volumes at lower spatial resolution, can be integrated into an efficient CMI pipeline bridging across scales from in vivo functional imaging experiments down to targeted vEM of selected regions in a large histologically resolved volume.

## Results

**Hard X-rays reveal subcellular context**. To evaluate the use of SXRT for imaging neural circuits, we used the mouse olfactory bulb (OB) as a model system[54]. The OB is a well-ordered layered structure (Fig. 1a-c) that contains information at different length scales: Each receptor neuron (ORN) in the nose expresses a single olfactory receptor, out of ~1000 possible[55,56]. ORNs extend their axons into the OB where those expressing the same type of receptor converge into 2–3 glomeruli, which are ~100 μm round neuropil structures. Beneath the layer of glomeruli there is a ~200 μm thick external plexiform layer (EPL), largely containing straight dendrites and sparse cell bodies, followed by a monolayer of large somata of projection neurons (mitral cell layer) and a densely populated granular inner layer. Furthermore, the OB is a superficial structure, accessible by in vivo functional imaging and it contains a number of diverse, prominent synaptic structures in the different layers. Altogether, the OB is a convenient model system to assess the limits of the different imaging techniques and their combination (Fig. 1a–d).

We thus stained mouse OB tissue sections with heavy metals using a protocol that preserves tissue ultrastructure[57] in preparation for X-ray imaging (Supplementary Fig. 1). As expected, a Zeiss Versa 510 laboratory-based micro-CT (LXRT) was readily capable of resolving individual glomeruli (Fig. 1e1, f1). While for example the thick layer of mitral cell somata could be identified with LXRT (Fig. 1g1), individual cell bodies were only rarely resolved (Fig. 1h1) and even the largest neurites remained undetected (Fig. 1g1).

In order to assess the benefit of high-brilliance, partially coherent X-rays, we imaged the same samples using SXRT at the beamline I13-2 at the Diamond Light Source[58] (Fig. 1e2–h2; Supplementary Fig. 2, see "Methods" for details on the SXRT methodology). Comparing the images of the same specimen obtained with both X-ray modalities revealed that SXRT provided a sufficient increase in spatial resolution to identify substructure

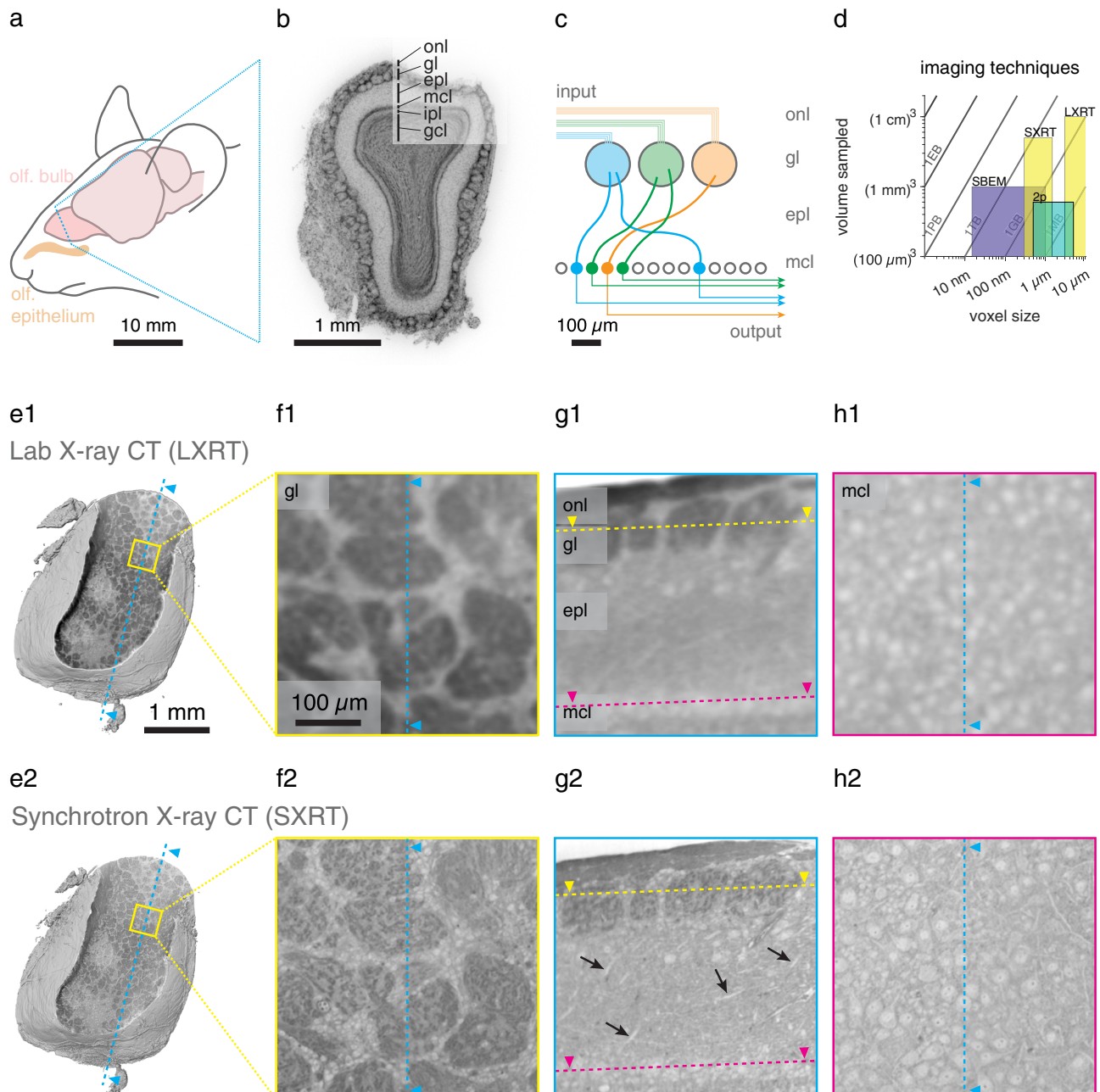

**Fig. 1 Correlative multimodal imaging for mouse brain tissue samples. a** Elements of the olfactory sensory pathway in the mouse brain. **b** Nuclear staining of a 100 µm coronal section of the mouse main olfactory bulb, displaying its histological layers, obtained with widefield fluorescence microscopy. **c** Diagram of the excitatory wiring of the glomerular columns in the olfactory bulb. **d** Imaging techniques capable of acquiring soft tissue volumes at the neural circuit scale. **e** Reconstruction of the LXRT (**e1**) and SXRT (**e2**) datasets obtained from the same sample, virtually sliced displaying the glomerular layer. **f–h** Virtual slices in three regions showing features resolved by each imaging modality in the glomerular layer (**f1-2**), in a sagittal section (**g1-2**) and in the mitral cell layer (**h1-2**). Arrows in (**g2**) indicate apical dendrites. Source data for (**d**) are provided as a Source Data file. onl, olfactory nerve layer; gl, glomerular layer; epl, external plexiform layer; mcl, mitral cell layer; ipl, inner plexiform layer; gcl, granule cell layer.

within each glomerulus (Fig. 1f2). Cell bodies of projection neurons (Fig. 1g2, h2) and smaller ones of juxtaglomerular cells (Fig. 1f2) were readily resolved in SXRT. Moreover, apical dendrites of mitral cells were recognisable in the SXRT datasets (black arrows, Fig. 1g2).

To quantify to what extent SXRT is capable of delineating dendrites in general, we used tissue samples from both hippocampus and olfactory bulb (Fig. 2, Supplementary Fig. 3). After SXRT (Fig. 2a, b, Supplementary Fig. 3g) we acquired serial block-face electron microscopy (SBEM) subvolumes at an

isotropic voxel size of 50 nm (olfactory bulb) and at $80*80*50$ nm$^3$ (hippocampus) (Fig. 2c, Supplementary Fig. 3h) as higher resolution "ground truth" data for assessing which features can be reliably identified in SXRT. For this, both EM and SXRT datasets of each region were warped into the same space[59], allowing us to identify the same neuronal features in each modality (Supplementary Fig. 4). Notably, as highly X-ray absorbing elements provided the strongest contrast in scanning EM (SEM) imaging as well[11,60], the patterns revealed by SXRT and SBEM were virtually identical (e.g. Fig. 2b, c), which

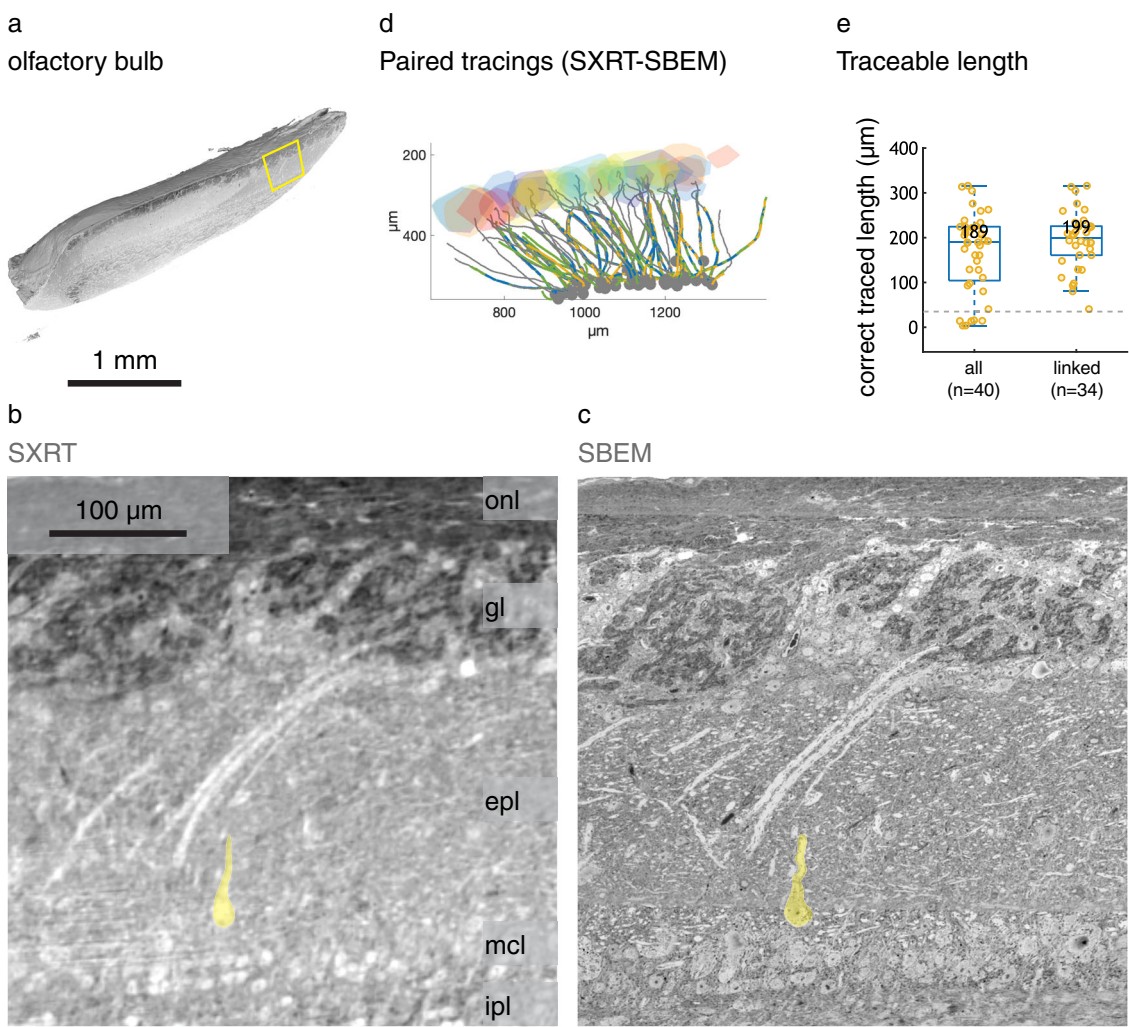

**Fig. 2 Apical dendrites of principal neurons can be traced in SXRT datasets. a** Volume of a mouse olfactory bulb obtained with SXRT, virtually sliced sagittally, displaying all main histological layers. Below, SXRT (**b**) and low-resolution SBEM ((50 nm)³ voxels) (**c**) of the highlighted region. **d** Apical dendrites traced in both imaging modalities by 3 independent tracers (grey lines: SBEM ground truth consensus, coloured lines: SXRT individual traces). **e** Traceable length of all SXRT dendrites and of the correctly linked ones. Traceable length measures the dendrite length through which SXRT tracing is within 12 µm (lost threshold) away from the paired EM tracing. Each dot represents one cell's apical dendrite. The box covers the 25 to 75% percentile range; the middle bar represents the median value (printed above); the whiskers extend to the most extreme data points that are not an outlier (defined as outside of the 1.5× interquartile range). The grey dashed line marks the lost threshold used. Source data for (**e**) are provided as a Source Data file. onl, olfactory nerve layer; gl, glomerular layer; epl, external plexiform layer; mcl, mitral cell layer; ipl, inner plexiform layer.

simplified warping procedures between these two modalities. We tagged the locations of the somata of 50 mitral and 132 pyramidal cells, and we asked 3 tracers to independently trace the apical dendrites of all those cells in both SXRT and SBEM as far as possible (Fig. 2d, Supplementary Fig. 3a, b, d–f). The consensus traces performed on SBEM data were used as ground truth, and independent tracers could trace dendrites in 44 cells on average. In the OB, the majority of apical dendrites were traced correctly in SXRT (29/44, averaged across three tracers, see "Methods") and could be followed for up to 369 µm, often sufficient to traverse the whole EPL (traceable length averaged among 3 tracers = 151.4 ± 24.6 µm, mean ± SEM) (Fig. 2d, e, Supplementary Fig. 3a, b).

Tracing apical dendrites was efficient across experiments (Supplementary Fig. 5) and reproducible on SXRT datasets from the same specimen acquired at two different parallel beam microtomography beamlines (I13-2 at the Diamond Light Source and TOMCAT at the Swiss Light Source) (Supplementary Fig. 3i–l). This suggests that dendrite-level anatomical data can

be efficiently achieved with many of the SXRT beamlines currently available at synchrotrons around the world. Most mistakes in following a dendrite's true trajectory (10/13, averaged across three tracers) happened at the region linking the soma and the apical dendrite (red lines in Supplementary Fig. 3a). In this region, the contrast between intracellular space and extracellular space is weaker than elsewhere in the dendrite, due to a higher density of endoplasmic reticula (Supplementary Fig. 6). Moreover, in a few cases (2/44) the tracers stopped tracing before that linkage region was solved (black lines in Supplementary Fig. 3a). Excluding the cells lost at the linkage region, the expected traceable length became both larger and less variable (average among 3 tracers = 201.8 ± 7.7 µm, mean ± SEM, Fig. 2e, Supplementary Fig. 3b), and indicative of the expected traceable length of apical dendritic processes alone in SXRT.

In the hippocampus, apical dendrites of pyramidal neurons could also be manually followed as they traversed the stratum radiatum, albeit with slightly shorter traceable length (mean 60.4 µm for all and 86.9 µm for linked cells, Supplementary

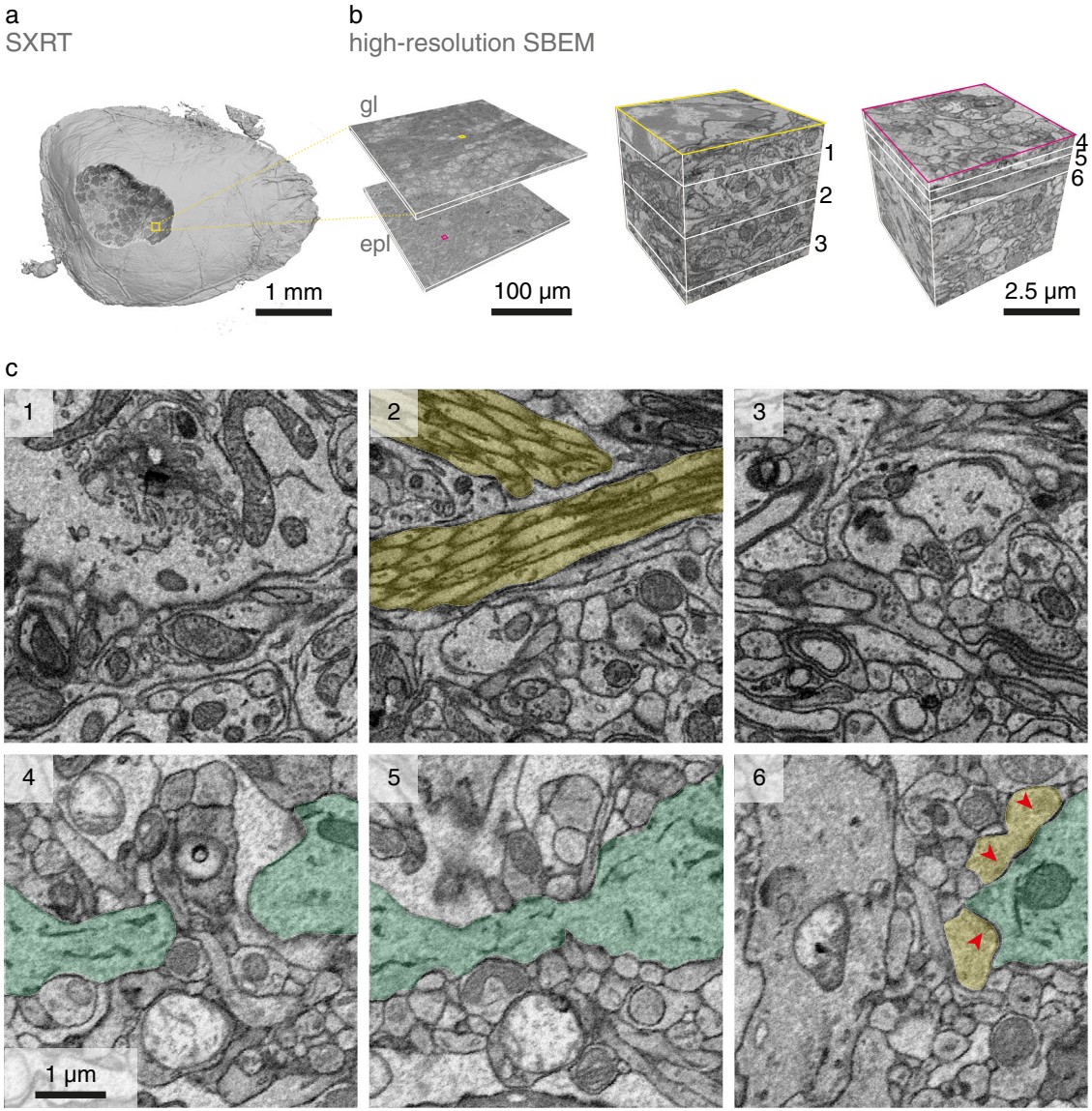

**Fig. 3 Tissue ultrastructure is preserved after SXRT.** Volumes obtained from the same sample: synchrotron X-ray CT (**a**) and posterior volume EM at high (10 × 10 × 40 nm³) resolution (**b**). Two regions of interest are cropped out to showcase the ultrastructure of the tissue. **c** Close-up details of the slices indicated in (**b**), displaying features indicative of well-preserved ultrastructure: axon bundles (yellow), thin dendrites (green) and synapses (red arrowheads). gl, glomerular layer; epl, external plexiform layer.

Fig. 3e, f), possibly due to the thinner diameter of those dendrites, the higher degree of similarity in the trajectories between dendrites compared to the OB, and variation in staining, amongst other factors. Altogether, these results indicated that, with appropriate contrast enhancing measures (e.g. staining), SXRT can be used to delineate subcellular features (e.g. neurites) reliably for several hundreds of micrometres.

Resolution estimates obtained with Fourier Shell Correlation were consistent with these findings (LXRT: 4 µm voxels, ~15 µm resolution; SXRT: 325 nm voxels; resolution of ~2-3 µm; SBEM: 50 nm voxels, resolution ~400 nm (Supplementary Fig. 7)).

**Multimodal imaging resolves brain anatomy across scales.** We next aimed to assess the extent of any radiation-induced damage inflicted upon SXRT-imaged samples, which has been reported to become a limiting factor for several high-resolution X-ray tomography imaging techniques[61–63]. Thus, we imaged the same samples after SXRT with high-resolution SBEM (Fig. 3, Supplementary Fig. 8). After radiating with ~10⁶ Gy using energies of

22 keV during SXRT, subsequent serial sectioning with a vibrating diamond knife[64] and SEM imaging were not noticeably impaired (Fig. 3b). Direct comparison between SBEM and SXRT demonstrated that the overall structure was unperturbed (e.g. Fig. 2b, c). To assess whether ultrastructure was maintained as well, we examined high-resolution EM images (10 nm pixel size, 40 nm section thickness) in selected regions of the OB (Fig. 3b, c). Ultrastructure was intact for fine axon bundles (Fig. 3c2), morphologies of cells and subcellular organelles in the glomerular layer (Fig. 3c1-3) as well as for apical dendrites (Fig. 3c4-6) and synaptic contacts in the EPL (Fig. 3c6).

Together this suggests that SXRT has the potential to be used to provide histological, subcellular context to subsequent targeted, high-resolution EM imaging. To assess the feasibility of combining and correlating the two imaging modalities, we turned to the mouse hippocampus, a heavily studied brain structure with clear layering and well-known synaptic circuitry. As expected, SXRT allowed us to identify pyramidal neuron cell bodies in the stratum pyramidale of CA1 (specifically subregion CA1a[65]), and

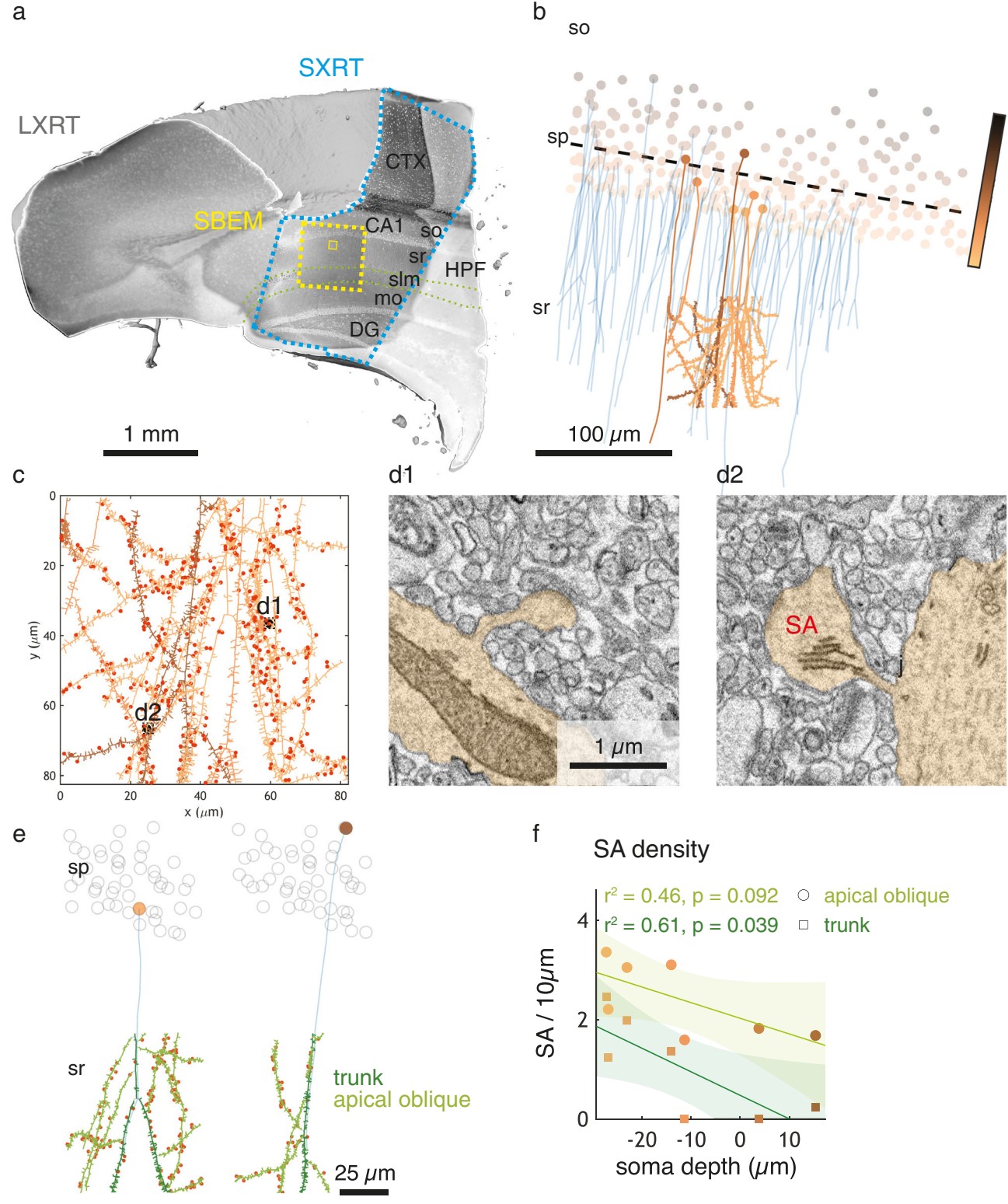

their apical dendrites could be traced through the stratum radiatum in SXRT (Fig. 4a, b, Supplementary Fig. 9). We targeted the medial stratum radiatum with SBEM to analyse dendritic spines on selected dendrites (Fig. 4a–d). We acquired a SBEM dataset of a volume of $82 \times 82 \times 39\ \mu m^3$ at a voxel size of $10 \times 10 \times 50\ nm^3$. We then traced in the SBEM volume the apical dendritic tree of 7 pyramidal neurons previously identified in

SXRT (Fig. 4b, c) and annotated all dendritic spines in those dendrites (adding up to a total of 3019 spines) (Fig. 4c). Of all spines 14% (417/3019) contained a spine apparatus (Fig. 4d), a specialised form of endoplasmic reticulum present in large spines and involved in calcium homoeostasis and synaptic plasticity[5,66–69]. As suggested previously[70], spines were denser on apical oblique dendrites than in trunk dendrites and the same

**Fig. 4 Multiscale analysis in the hippocampal CA1a region using correlative LXRT, SXRT and SBEM. a** Hippocampus acquired with LXRT, SXRT (blue dashed line) and SBEM (yellow lines, dashed: low resolution; solid: high resolution). **b** CA1 pyramidal neurons. Soma colours refer to their position in the pyramidal layer. Apical dendrites (traced in the SXRT dataset) are shown in blue. The dashed line shows the average depth of all seeded somata. The dendrites of seven cells were traced exhaustively in SBEM (shown in the same colour as their soma). **c** Dendrites and spines traced in SBEM. Spines containing a spine apparatus are highlighted in red. **d** Examples of spines with (d2) and without (d1) spine apparatus. **e** CA1 pyramidal neurons. The apical dendrite traced in the SXRT dataset is shown in blue, dendrites traced in SBEM are shown in green (dark: trunk; light: apical oblique). Spines are shown as small protrusions, a red mark highlighting those with a spine apparatus. **f** Density of spine apparati. $n = 7$ cells from one dataset, linear regression, $p = 0.092$, 0.039 for apical oblique and trunk dendrites, respectively. The shaded area represents the 95% confidence interval of the linear regression line. Source data for (**f**) are provided as a Source Data file. CTX, cerebral cortex; HPF, hippocampal formation; CA1, Ammon's horn field CA1; so, stratum oriens; sp, stratum pyramidale; sr, stratum radiatum; mo, molecular layers; DG, dentate gyrus.

applied to spines containing spine apparatus (Supplementary Fig. 9).

Hippocampal pyramidal cells have been proposed to differ according to soma position (deeper or more superficial in the pyramidal cell layer). To assess whether spine properties depend on soma depth, we linked the traces of apical dendrites in the high-resolution SBEM volume with those in the SXRT dataset, reaching the cell body layer (Fig. 4b). This allowed us to give each dendritic branch and spine their specific neural "depth" identity based on the position of their soma within the thickness of the pyramidal layer (i.e. more superficial or deeper). Spine density was independent of soma position (Supplementary Fig. 9), but more superficial pyramidal cells indeed had a substantially larger density of spines containing spine apparatus (Fig. 4e, f). This supports the notion that some properties of CA1 pyramidal neurons may systematically vary between neurons, depending on soma depth within the CA1 pyramidal layer[65,71,72]. More generally, this experiment showed how SXRT can provide context —cellular identity— to an EM-powered analysis of tissue ultrastructure.

We next aimed to corroborate how SXRT imaging can benefit the wider neuroscience community. Thus, we imaged a number of other brain regions with SXRT, including the mouse neocortex, striatum, and cerebellum (Fig. 5). Again, cell bodies, neurites and axon bundles were all readily visible in SXRT datasets of all brain regions, emphasising the versatility of the approach. This suggests that SXRT can generally provide subcellular context across multiple mm³ of soft tissue non-destructively. This in turn can be used to inform follow-up higher resolution imaging of smaller targeted regions of interest employing compatible techniques such as vEM.

**Bridging in vivo function with ex vivo structure.** One of the key challenges in mammalian circuit neuroscience is to bring together large-scale functional studies in vivo with their structural anatomical underpinning[8,37,73]. Full neural circuits are embedded in volumes often exceeding several mm³ [3]. However, acquiring synaptic resolution datasets of volumes nearing a 1 mm³ using vEM has only been achieved in individual exceptional circumstances[37,74]. Targeting more readily obtainable high-resolution ~(10–100 μm)³ volumes acquired with vEM to specific locations within mm³ volumes previously imaged in vivo is typically highly challenging and error-prone. The different length scales and staining protocols associated with the different imaging modalities of this multi-step process often result in structural distortions that need to be untangled to robustly correlate features across datasets. Thus, we hypothesised that the large volumes and high level of structural detail provided by SXRT should substantially simplify this CMI pipeline.

To assess this we aimed to directly link in vivo 2-photon Ca²⁺ functional data to underlying anatomical features in the mouse OB circuitry. We presented a panel of 48 distinct odours to anaesthetised mice[75] and recorded the activity of projection

neurons (mitral and tufted cells) across a glomerular column[76]. We narrowed our analysis to the glomerular layer, where the dendrites of projection neurons delineate glomerular contours. We collected the averaged odour-evoked response profiles of 20 glomeruli on the dorsal surface of the OB (Fig. 6a–d). Following in vivo functional imaging, we labelled the blood vessels by injecting the fluorophore sulforhodamine (SR101) intraperitoneally and acquired in the same in vivo 2-photon setup a fluorescence dataset containing the entire tissue volume with higher spatial resolution (~0.36 μm horizontal and 5 μm vertical spatial voxel size). Mice were then sacrificed, the OB extracted and a 600 μm dorsal section including all imaged glomeruli dissected out. Imaging the tissue section in the 2-photon microscope after fixation ('2p ex vivo') provided a dataset where glutaraldehyde-induced autofluorescence delineates glomeruli alongside numerous histological landmarks. Subsequently, tissue was stained as described above and imaged with LXRT, SXRT, and finally – after trimming to a tissue block of suitable dimensions (C525a: 930 × 1080 × 630 μm, C525b: 970 × 960 × 590 μm, wide × long × tall)—in SBEM (Supplementary Fig. 8). Histological landmarks from glutaraldehyde autofluorescence were used to warp the 2p ex vivo and the LXRT datasets, while the 2p in vivo anatomical dataset was warped to the SXRT and SBEM datasets directly using the unique patterns of the blood vessel network as landmarks (Supplementary Fig. 4).

This CMI pipeline is efficient enough to allow routinely identifying genetically tagged glomeruli (Supplementary Fig. 5) as well as targeting several mm³-large features in vivo in the same specimen (e.g. different glomerular columns, Fig. 7) with subsequent SBEM (Fig. 7, Supplementary Fig. 10). As before, laboratory- and synchrotron-based X-ray imaging and EM revealed different levels of detail of the glomerular anatomy, with SXRT resolving essentially all glomeruli in the field of view (as gauged by EM acquisition of a targeted subvolume, Fig. 6e–h). A first readout of this correlated experiment provided a matched census of biological features identified by the different imaging modalities. It corroborated that the majority of functionally recorded regions of interest (ROIs) (17/20) corresponded to independent glomeruli, and provided insights into the biological substrates from which the remaining ROIs (3/20) were collecting the signal. This ratio informs the extent to which only X-ray or EM datasets might correctly attribute fluorescence to anatomical units (Fig. 6i, j, Supplementary Figs. 10, 11).

## Discussion

Here we have shown that synchrotron X-ray computed tomography with propagation-based phase contrast (SXRT) can be used to efficiently link 2-photon in vivo imaging to targeted vEM by bridging the gap between the length scales in which those two modalities operate. SXRT, which provides superior spatial resolution compared to standard laboratory X-ray μCTs (LXRT) and thus more detailed and precise anatomical landmarks, makes this correlative multimodal imaging (CMI) pipeline efficient and reliable (e.g. Fig. 7, Supplementary Fig. 4). We suggest terming

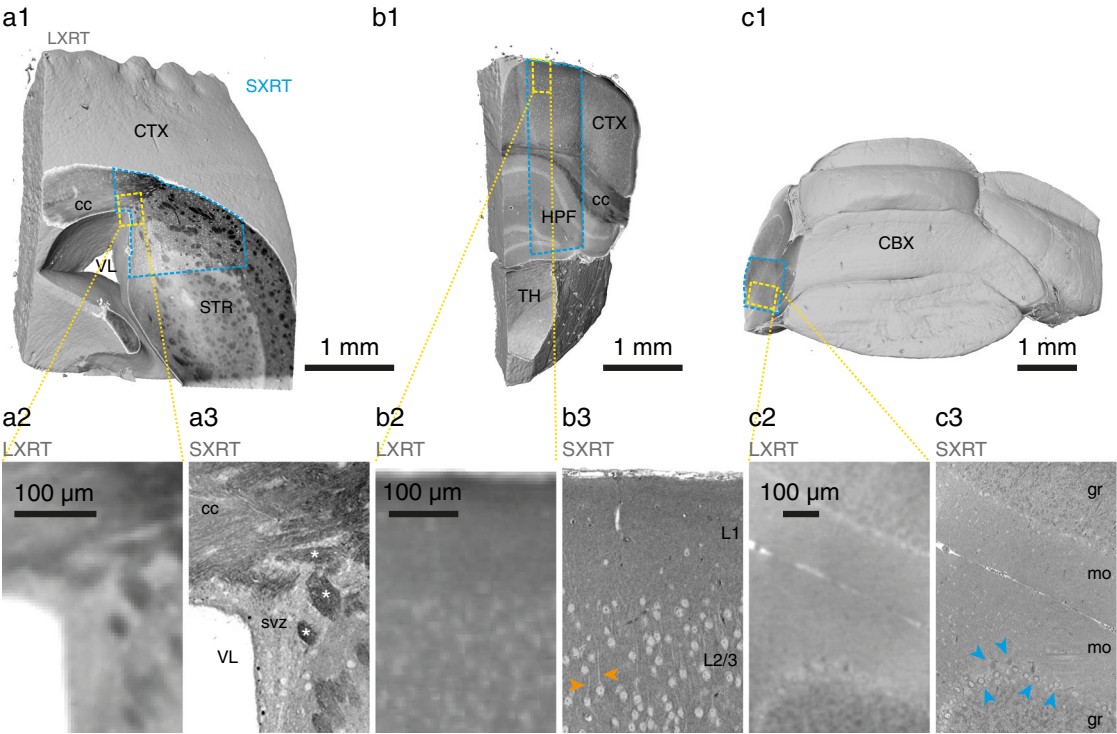

**Fig. 5 Subcellular compartments resolved by SXRT in other mouse brain regions.** Full-sample panels (**a1**, **b1**, **c1**) display SXRT data where available (boundaries in blue dashed lines) and LXRT data elsewhere. Detailed panels show LXRT (**a2**, **b2**, **c2**) and SXRT (**a3**, **b3**, **c3**) data from the regions highlighted by yellow dashed lines in (**a1**, **b1**, **c1**), respectively. (**a**) Mouse brain sample containing striatum, corpus callosum and cortex. Detail (**a3**) shows the subventricular zone, somata and myelinated axon bundles (white asterisks). **b** Sample containing cerebral cortex, hippocampus and thalamus. Detail (**b3**) shows cortical layers 1 and 2/3, displaying somata and apical dendrites of pyramidal neurons (orange arrowheads). **c** Sample containing cerebellar cortex. Detail (**c3**) shows a section across cerebellar lobules, displaying granular and molecular layers and purkinje neurons (blue arrowheads). CTX, cerebral cortex; STR, striatum (caudoputamen); VL, lateral ventricle; cc, corpus callosum; svz, subventricular zone; white asterisks, myelinated axon bundles; HPF, hippocampal formation; TH, thalamus; L1, cortex layer 1; L2/3, cortex layer 2/3; Orange arrowheads, apical dendrites of cortical pyramidal neurons; CBX, cerebellar cortex; gr, granular layer; mo, molecular layer; blue arrowheads, Purkinje neurons.

this use of X-ray microscopy, "bridging use"[10]. Moreover, we have demonstrated that SXRT provides sufficient resolution to delineate individual cell bodies and at least some neurites in a dense fashion. This information on the subcellular context over several millimetres can be quantified alongside analyses relying on targeted high-resolution vEM measurements. Thus, SXRT not only provides bridging information but can also be employed for "context use"[10].

We have applied this CMI pipeline in two mammalian neuronal circuits. We have shown that the trunk dendrites of more superficial CA1a pyramidal neurons display more spines containing spine apparatus in the stratum radiatum than neighbouring dendrites of deeper cells. Deep and superficial CA1 pyramidal neurons have been shown to display specific patterns of both connectivity and spine density in their dendritic tufts at the stratum lacunosum-moleculare (slm)[65]. In CA1a, more superficial neurons receive stronger input from the lateral entorhinal cortex into their tufts at slm and also display a larger density of spines in those dendritic tufts at slm[65]. Here we have shown that their more proximal dendritic trunks and apical oblique dendrites at stratum radiatum also have a higher density of spines containing spine apparatus, a specialised form of endoplasmic reticulum involved in the regulation of the dynamics of calcium transients[5,66–69]. This suggests that calcium transients in the spines might follow distinct dynamics, differing between dendrites belonging to neurons with deeper and more superficial somata. This supports the general notion that the radial axis in the CA1 pyramidal layer could explain some of the variability in the computations performed by different pyramidal neurons. Spine apparati are difficult to observe

without high-resolution EM, yet spine morphology and local dendritic morphology do not obviously differ between different pyramidal neurons. Thus, to identify a difference in the spine apparatus distribution between superficial and deeper pyramidal cells required combining high-resolution EM analysis with context information provided by correlative SXRT.

Furthermore, we have shown in the OB that functional $Ca^{2+}$ imaging in vivo can be combined with SXRT and SBEM (Fig. 7), and that this CMI pipeline can be used to delineate the anatomical regions—glomeruli—where functional imaging data has been recorded. This was more accurate than by 2p imaging alone and can help to disambiguate signal sources. While glomeruli are ~100 μm in diameter, the resolution of SXRT will allow correlating smaller regions reliably, possibly down to individual cells. The tools developed here will then allow to transform annotations (structural ones on different length scales such as subcellular or regional segmentation as well as e.g. functional signals) from the modality with the highest resolution to the corresponding location in the other modalities and vice versa. For example, neurites traced in EM can be visualised in and linked up with other circuit elements in the SXRT space, or neurons outlined in SXRT can be attributed functional signals from an in vivo 2p dataset.

To allow the combination of in vivo imaging with LXRT, SXRT and SBEM it was critical to obtain sufficient contrast with all modalities while preserving sample integrity. This constraint had different implications on each imaging step, imposed the need for developing accurate quality controls between them, and shed light on how the different modalities could in the future provide better data quality. When imaging with X-rays, this constraint implies

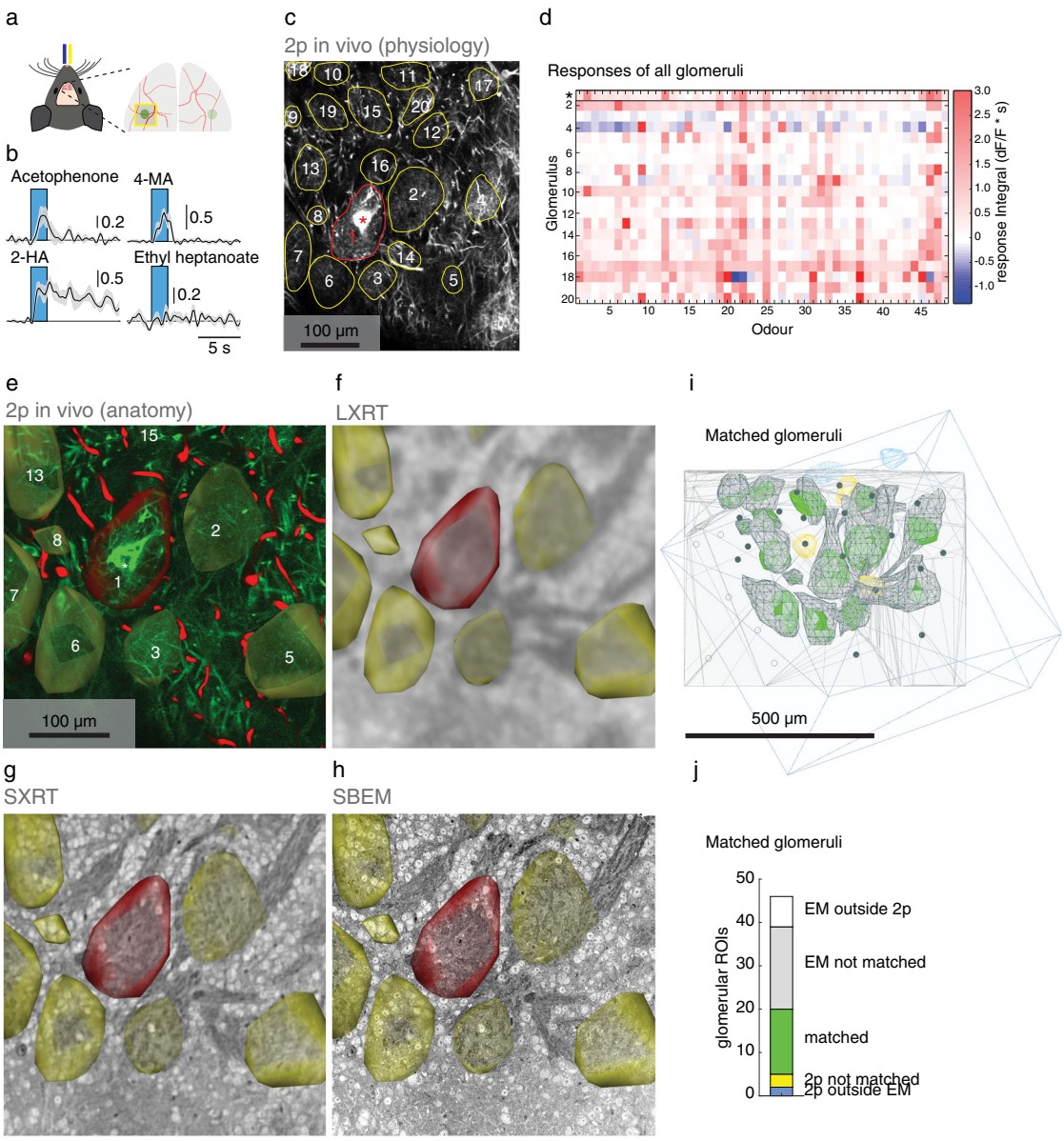

**Fig. 6 Multiscale analysis in the olfactory bulb glomerular layer using correlative in vivo 2-photon Ca$^{2+}$ imaging, LXRT, SXRT and SBEM. a** in vivo 2-photon imaging setup for imaging neural activity of projection neurons in anaesthetised mice in response to a controlled odour pattern delivered to their nose (blue and yellow ports indicate odour delivery and airflow sensor). Right: Schematic of the olfactory bulb surface with blood vessels (red) and a genetically targeted glomerulus (green). **b** Example traces (mean of 5 presentations ± SEM) of neuronal activity in response to four odours (odour presentation window in blue). **c** Maximum projection (8000 frames) of a functionally imaged plane recorded in vivo, located in the glomerular layer. Highly branched dendritic tufts define the contours of 20 putative glomeruli (yellow). The fluorescence signal inside these regions of interest was measured over time. In the M72 glomerulus the afferent sensory axons expressed YFP, making it identifiable (red asterisk and contour). **d** Response integral of all glomeruli to all odours (average of 5 presentations). **e** Dataset obtained in vivo at the end of the functional recordings using 2-photon microscopy. (green: GCaMP/M72-YFP; red: blood vessels labelled by SR101). This dataset was warped into the same common space as (**f–h**). **f** Same region as in (**e**), as obtained by LXRT. **g** Same region as in (**e**), as obtained by SXRT. **h** Same region as in (**e**), as obtained by SBEM. **i** Extents of the 2p anatomical (blue box) and SBEM datasets (pale grey), all functionally imaged ROIs, coloured based on whether they were found inside a glomerulus in EM (green), outside the EM-imaged volume (blue) or located outside of any glomerulus despite being inside the EM-imaged volume (yellow). Detailed reconstructions of the matched EM glomeruli are shown in grey. Centroids of all other glomeruli in the EM dataset that are also in the 2p-imaged volume are shown as filled dots. Centroids of glomeruli in the EM dataset located outside the 2p-imaged volume are shown as empty dots. **j** Yield of the correlative experiment (number of glomeruli matched across imaging modalities). Source data for (**b**, **d**, **i**, **j**) are provided as a Source Data file.

avoiding inflicting radiation damage. Since obtaining higher resolution with X-ray techniques might require higher radiation doses (compared to the ~10$^6$ Gy used here), mechanisms to attenuate radiation damage such as cryogenic cooling during data collection might become necessary[48]. A key challenge for vEM has been to reach sufficient contrast for low current (~200 nA),

high-speed imaging (2 MHz), which has been overcome by en-bloc staining protocols rich in heavy-metal content. We found that standard osmium/uranium/lead en-bloc EM staining[57] is not only compatible with SXRT but moreover provides excellent contrast for sample depths of up to ~600 µm. Improvements in this staining protocol[57,77,78] as well as in making imaging both

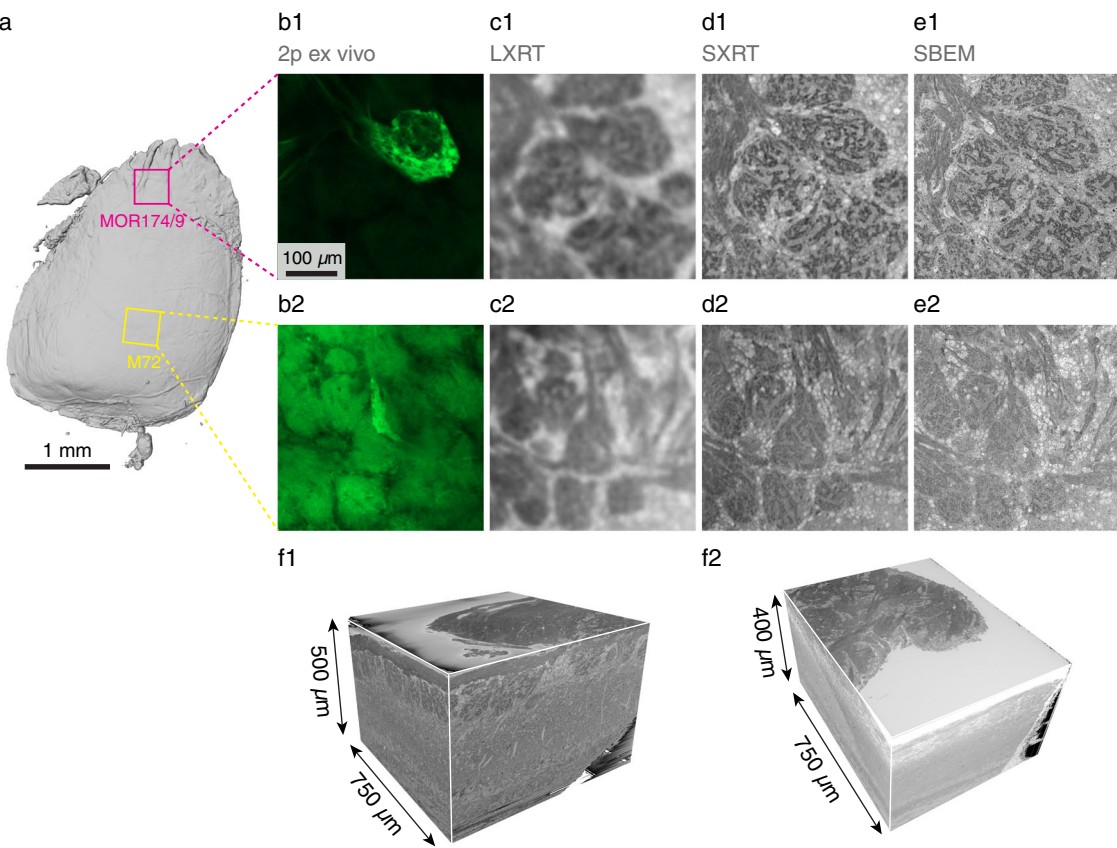

**Fig. 7 Multiple SBEM datasets obtained of two regions from the same specimen targeted using correlative ex vivo 2-photon, LXRT, SXRT and SBEM.** (**a**) Reconstruction of an olfactory bulb slab, displaying the regions of interest containing the MOR174/9 and M72 glomeruli in magenta and yellow, respectively. (**b–e**) Multimodal correlated image data of the same region, displaying the 2p ex vivo dataset of the fixed specimen (**b**), LXRT (**c**), SXRT (**d**) and SBEM (**e**) of the MOR174/9 (**b1–e1**) and M72 (**b2–e2**) regions of interest delineated in (**a**). **f** Reconstructions of the SBEM datasets containing both genetically identified glomerular columns.

more sensitive and more robust (e.g. by reducing its overall vulnerability to charge build-up-derived artefacts using e.g. focal charge compensation mechanisms[30]) will enable thicker samples to be efficiently stained and systematically imaged, and enhance the overall yield when combined with prior in vivo imaging.

The context use of SXRT has also highlighted the potential benefits of using this technique to directly retrieve biological insight. If compatibility with vEM is not a requirement, it is possible that dedicated sample staining protocols for SXRT imaging might further improve resolution or contrast. X-ray contrast in SXRT can be generated by two mechanisms: absorption and phase shift[79]. Staining the protein and lipid distribution in the samples (i.e. their ultrastructure) by passively diffusing heavy metals through the tissue exploits the former mechanism —heavier atoms absorbing more X-rays— but also limits the thickness a sample can have before becoming opaque to the beam. Lighter metal staining combined with X-ray phase-contrast imaging would enable imaging larger tissue blocks with SXRT at a similar energy. An alternative approach would be to rely on phase shift alone to generate contrast, in which case the ultrastructure would have to be preserved while minimising the use of highly absorbing heavy metals (e.g. by employing only trace concentrations of Osmium or heavily scattering lighter elements). This could also allow the use of a wider range of phase retrieval algorithms[80]. Alternatively, thicker specimens could be imaged at higher energies. Here, new and upcoming fourth-generation synchrotron sources[81–84] offer substantially increased coherent photon flux at high energies and, together with technical advances in e.g. undulator technology will likely make tomography at synchrotrons a rapid, high-throughput technique. Recently developed high-performance liquid metal jet X-ray sources in turn have allowed phase-contrast imaging outside of synchrotrons[39,85] and could provide another alternative for rapid large volume X-ray imaging at micrometre resolution for broad use. Finally, all these developments are likely to not only improve the yield of SXRT imaging, but also that of other promising hard X-ray imaging techniques compatible with soft biological tissues, such as nano-holotomography[48] (Supplementary Fig. 13) or ptychography[49,51,86]. While at this point these techniques require smaller samples than the full-field parallel beam microtomography approaches described here, they have the potential of achieving resolutions of tens of nanometres.

We have described a correlative multimodal imaging pipeline capable of bringing together insights on the function and structure of mammalian neural circuits embedded in mm$^3$ volumes. It combines temporal and spatial resolution of in vivo 2-photon Ca$^{2+}$ imaging with both multi-mm$^3$ structural maps at subcellular detail given by SXRT and with synaptic resolution detail provided by SBEM at targeted locations. It provides a 'bridging use' of SXRT to target SBEM to optimal locations with a 'context use' of SXRT (providing information about neurites and somata) that has biological value on its own. We have employed this pipeline to investigate two mammalian neuronal circuits, and we report its applicability to other brain regions. We are confident this approach will be a powerful tool to investigate structure and function of neuronal circuits in mammalian brains. Moreover, by

introducing a routine way to bridge the gap in length scales between organs and subcellular structures, it will benefit tissue biology more widely.

## Methods
All samples, animal IDs and corresponding figures are listed in Supplementary Table 1.

**Animals**. Animals in this study were all 8–13-week-old mice. For experiment C525, we used a male transgenic mouse resulting of MOR174/9-eGFP[87] crossed into M72-IRES-ChR2-YFP[88] (JAX stock #021206) crossed into a Tbet-cre driver line[89] (JAX stock #024507) crossed with a GCaMP6f reporter line[90] (JAX stock #028865). For experiment C556, we used a male mouse of C57Bl/6 background. For the rest of the experiments, we used mice of either gender of C57Bl/6 background. All animal protocols were approved by the Ethics Committee of the board of the Francis Crick Institute and the United Kingdom Home Office under the Animals (Scientific Procedures) Act 1986.

**Statistics and reproducibility**. All images displaying biological features report intuitive examples of features robustly resolved in independent specimens. Specimens from which all panels were extracted are detailed in Supplementary Table 1.

**In vivo imaging**
*Surgical and experimental procedures*. Prior to surgery all utilised surfaces and apparatus were sterilised with 1% trigene. Mice aged 10 weeks were anaesthetised using a mixture of fentanyl/midazolam/medetomidine (0.05, 5, 0.5 mg/kg respectively). Depth of anaesthesia was monitored throughout the procedure by testing the toe-pinch reflex. The fur over the skull and at the base of the neck was shaved away and the skin cleaned with 1% chlorhexidine scrub. Mice were then placed on a thermoregulator (DC Temperature Controller, FHC, ME USA) heat pad controlled by a temperature probe inserted rectally. While on the heat pad, the head of the animal was held in place with a set of ear bars. The scalp was incised and pulled away from the skull with four arterial clamps at each corner of the incision. A custom head-fixation implant was attached to the base of the skull with medical super glue (Vetbond, 3 M, Maplewood MN, USA) such that its most anterior point rested approximately 0.5 mm posterior to the bregma line. Dental cement (Paladur, Heraeus Kulzer GmbH, Hanau, Germany; Simplex Rapid Liquid, Associated Dental Products Ltd., Swindon, UK) was then applied around the edges of the implant to ensure firm adhesion to the skull. A craniotomy over the left olfactory bulb (approximately 2 × 2 mm) was made with a dental drill (Success 40, Osada, Tokyo, Japan) and then immersed in ACSF (NaCl (125 mM), KCl (5 mM), HEPES (10 mM), pH adjusted to 7.4 with NaOH, MgSO4.7H2O (2 mM), CaCl2.2H2O (2 mM), glucose (10 mM)) before removing the skull with forceps. The dura was then peeled back using fine forceps. A layer of 2% low-melt agarose diluted in ACSF was applied over the exposed brain surface before placing a glass window cut from a cover slip (borosilicate glass #1 thickness [150 μm]) using a diamond scalpel (Sigma-Aldrich) over the craniotomy. The edges of the window were then glued with medical super glue (Vetbond, 3 M, Maplewood MN, USA) to the skull.

Following surgery, mice were placed in a custom head-fixation apparatus and transferred to a two-photon microscope rig along with the heat pad. The microscope (Scientifica Multiphoton VivoScope) was coupled with a MaiTai DeepSee laser (Spectra Physics, Santa Clara, CA) tuned to 940 nm (<30 mW average power on the sample) for imaging. Images (512 × 512 pixels, field of view 550 × 550 μm) were acquired with a resonant scanner at a frame rate of 30 Hz using a 16 × 0.8 NA water-immersion objective (Nikon). Using a piezo motor (PI Instruments, UK) connected to the objective, a volume of ~300 μm was divided into 12 planes resulting in an effective volume repetition rate of ~2.5 Hz. The odour port was adjusted to approximately 1 cm away from the ipsilateral nostril to the imaging window, and a flow sensor (A3100, Honeywell, NC, USA) was placed in the contralateral nostril for continuous respiration recording and digitised with a Power 1401 ADC board (CED, Cambridge, UK).

Following odour stimulations (see below), Sulforhodamine 101 (Sigma Aldrich, 100 μM final concentration) was injected intraperitoneally to label blood vessels and a z stack of the entire volume was acquired, consisting of 88 images each covering $(464 \, \mu m)^2$ with $(358 \, nm)^2$ pixels, separated in z by 5 μm.

*Odour stimulation*. Odour stimuli were delivered using a set of custom-made 6-channel airflow dilution olfactometers. In brief, volumes of 3 ml of a set of 48 monomolecular odorants (Sigma-Aldrich, St. Louis MO, USA) were pipetted freshly for each experimental day into 15 ml glass vials (27160-U, Sigma-Aldrich, St. Louis MO, USA). Odours were diluted 1:20 in air before being presented to the animal at 0.3 litres/min using custom Python Software (PulseBoy; github.com/RoboDoig). Odours were prepared for 3 sec in tubing before a final odour valve was triggered to open at the beginning of an inhalation cycle. During anaesthetised recordings all stimuli were presented with a 30 s inter-stimulus interval. To minimise contamination between odour presentations, a high flow clean air stream was passed through the olfactometer manifolds during this time.

*Data analysis*. Motion correction, segmentation and trace extraction were performed using the Suite2p package (https://github.com/MouseLand/suite2p). Regions of Interest (ROIs) corresponding to glomeruli were manually delineated based on the mean fluorescence image in Fiji[91]. The fluorescence signal from all pixels within each ROI was averaged and extracted as a time series with $\Delta F/F = (F - F_0)/F_0$, where $F$ = raw fluorescence and $F_0$ = median of the fluorescence signal distribution. Further analysis was performed with custom Matlab scripts.

**Tissue preparation**
*Dissection*. Mice were sacrificed and 600 μm thick sections of brain areas of interest were sliced in ice-cold dissecting buffer (phosphate buffer 65 mM, 0.6 mM CaCl2, 150 mM sucrose) with an osmolarity of 300 ± 10 mOsm/L using a LeicaVT1200S vibratome and immediately transferred to ice-cold fixative (2% glutaraldehyde in 150 mM sodium cacodylate buffer pH 7.40, 300 mOsm/L). A stereoscope (Leica MZ10 F) was mounted above the vibratome to guide slicing and confirm presence of fluorescent landmarks on the fresh sections when present. At this point, the olfactory bulb (OB) section was quickly imaged using a widefield fluorescence microscope (Zeiss Axioplan2), recording the coarse location of the fluorescently tagged glomeruli M72-YFP and MOR174/9-eGFP. Samples were left in the same fixative overnight, at 4 °C. The fixative was then washed with wash buffer (150 mM sodium cacodylate pH 7.40, 300 mOsm/L) three times for 10 min at 4 °C. The OB section was then imaged at the 2-photon microscope in the locations surrounding both tagged glomeruli to record the 3D anatomical background autofluorescence generated by glutaraldehyde. Overall, samples were kept in an ice-cold, osmolarity-checked buffer.

*Staining, dehydration and embedding*. Slabs were stained with heavy metals using an established ROTO protocol[57] using an automated tissue processor (Leica EMTP). Briefly, they were first stained with reduced osmium (2% OsO4, 3% potassium ferrocyanide, 2 mM CaCl2 in wash buffer) for 2 h at 20 °C, followed with 1% thiocarbohydrazide (aq) at 60 °C for 50 min, 2% osmium (aq) for 2 h at 20 °C, and 1% uranyl acetate (aq) overnight at 4 °C. On the next day, the samples were further stained with lead aspartate (prepared as in[92]) for 2 h at 60 °C. Samples were washed with double-distilled water six times for 10 min at 20 °C between each staining step, except warmer washes before and after TCH (50 °C) and before LA (60 °C).

Samples were then dehydrated with increasing ethanol solutions (75, 90, 2 × 100%), transferred to propylene oxide, and infiltrated with hard epon[93] mixed with propylene oxide in increasing concentrations (25, 50, 75, 2 × 100%). Finally, samples were polymerised individually into plastic moulds for 72 h at 70 °C. While final chemical composition is difficult to estimate[60], overall density of the polymerised resin was 1.24 ± 0.01 g/ml (n = 15 blocks of resin).

**SXRT imaging**. Experimental parameters for synchrotron X-ray imaging are listed in Supplementary Table 2.

Samples were imaged with full-field X-ray tomography at the I13-2 beamline of the Diamond synchrotron (Didcot, UK) using a polychromatic X-ray beam from an undulator source, filtered by 1.34 mm pyrolytic graphite, 2.1 mm aluminium, 0.035 mm silver, and 0.042 mm palladium. This produced an X-ray spectrum with an effective energy of approximately 22 keV. Samples were mounted at a distance of 52 mm upstream of the detector system. The latter consisted of a scintillation screen (34 μm-thick Europium-doped Gadolinium Gallium Garnet (GGG:Eu)), a 10x Olympus UPlan S Apo objective lens, 2x tube lens and a pco.edge 5.5 camera (sCMOS chip with 2560 × 2160 pixels). This system yielded an effective pixel size of 325 nm. The choice of objective was dictated by availability and a compromise between acquisition time and final image resolution (see also Supplementary Fig. 14 for a comparison of 20x and 40x objectives at PSI TOMCAT).

A set of 2–24 local tomography scans (with an overlap of 0.22 mm in the horizontal and 0.1 mm in the vertical direction) was acquired for each specimen to cover a total volume of 1.2–9.9 mm³ (Supplementary Fig. 1b). For each tomography, 3001 projections were recorded at equi-distant viewing angles over 180° of sample rotation with an exposure time of 0.4 s per frame, which resulted in a total scan time of approximately 20 min per tomography dataset.

All data were saved in hdf5 container files. We used the reconstruction pipeline *savu*[94] for reconstructing the SRXT datasets. The full processing pipeline steps were: a. Loading of HDF5 dataset; b. Correction for dark and flat-field images; c. Paganin filter; d. Ring removal; e. Automatic centre-finding; f. Reconstruction; g. Saving images as tiff files. Dark images, flat-field images and projections were extracted from the HDF5 dataset (a.) and all projections $p$ were corrected for detector dark current $d$ and the flat-field $f$ intensity variation (b.). The normalised projections pnorm were calculated using: $p_{norm} = \frac{p-d}{f-d}$. The normalized projections were then filtered with a Paganin filter (also see below)[95] using the following settings: detector distance 52 mm, X-ray energy 22 keV and detector pixel size 325 nm. Afterwards, processing followed the algorithms outlined in[96]. In brief, ring artefacts were detected using algorithm #4 (from[96]) and rings were removed using algorithms #6, #5, and #3 (from[96]). For algorithms #6 and #5, a signal-to-noise ratio (SNR) of 3.0 and a filter window size of 71 pixels was used. The window size used for algorithm #3 was 31 pixels. The centre of reconstruction was determined as described in[96] and the reconstruction was performed using the GPU

implementation of the Astra toolbox[97–99]. A filtered back-projection algorithm with a Ram-Lak filter and outer padding (padding factor of $\sqrt{2}$) was applied to the Paganin-filtered, normalized projections. The resulting data was saved as 32-bit tiff images.

One sample (C432) was subsequently imaged with full-field X-ray tomography at the TOMCAT beamline of the Swiss Light Source at the Paul Scherrer Institut (Villigen, Switzerland) using a multilayer monochromatic X-ray beam filtered by 0.1 mm aluminium and 0.01 mm iron. This produced an X-ray spectrum with an effective energy of 21 keV. Samples were mounted at a distance of 50 mm upstream of the detector, consisting of a scintillation screen (LuAG:Ce 20 μm), a 20x Olympus UPlan Apo objective lens and a pco.edge 5.5 camera, yielding to an effective pixel size of 325 nm. 24 tomography scans (with an overlap of 0.25 mm in the horizontal and 0.03 mm in the vertical direction) were acquired covering the entire specimen (8.9 mm³). For each tomography, 2601 projections were recorded over 180° of sample rotation with an exposure time of 0.3 s per frame, which resulted in a total scan time of approximately 13 min per tomogram. The tomograms were reconstructed using the same protocol as above. The parameters used for the single-distance phase retrieval were put to delta = 2e−6 and beta = 6e−7. An unsharp mask filter was applied with a stabiliser of 0.6 and a Gaussian kernel filter of 1.0. The tomographic reconstruction was performed using the GridRec algorithms[100].

The different volumes obtained by the local tomographies were stitched together using a 3D non-rigid stitching algorithm based on local pairwise cross-correlation[101].

In order to probe image reconstruction for different reconstruction algorithms (see below) and under conditions of varying beam coherence, two samples were further imaged at the microtomography beamline ID19 and the nano-holotomography beamline ID16A at the European Synchrotron Radiation Facility (ESRF) in Grenoble (Supplementary Figs. 12, 13, Supplementary Table 2).

*Phase retrieval.* Phase tomograms were reconstructed by applying a commonly used single-distance phase retrieval algorithm[95] to the 2D projections and subsequently performing the tomographic reconstruction using a filtered back-projection algorithm. For the single-distance phase retrieval, unless noted differently, a ratio of the real part delta and imaginary part beta of the refractive index was set to delta/beta = 1, based on visual optimisation of the reconstructed images (Supplementary Fig. 15). While the heavy-metal stain employed results in significant X-ray absorption, transmission was still sufficient for efficient phase retrieval (40 ± 4% transmission through the short [0.6 mm] axis [range 31–51%] and 7 ± 1% through the long [3 mm] axis [range 4–12%] of the sample). Notably, Paganin-based phase retrieval as employed here visually improved image reconstruction compared to the pure absorption regime (Supplementary Fig. 15). While in principle Paganin phase retrieval algorithms assume sample homogeneity it has been argued that in practise they are rather insensitive to inhomogeneities as long as elemental composition is homogeneous and varies only in density[102]. Moreover, in the case of heavy-metal stained brain tissue, samples are in fact rather homogeneous on the 100–1000 nm length scale (Supplementary Fig. 16).

Experimental parameters for the different experiments described are listed in (Supplementary Table 2). Notably, Fresnel numbers for the microtomography beamlines I13-2 and TOMCAT are <1 indicating that other phase retrieval approaches such as the Contrast-Transfer-Function (CTF) method[80,103] might be appropriate. While for high beam coherence CTF provides higher resolution compared to Paganin phase retrieval (Supplementary Fig. 12), for microtomography beamlines optimised for high flux, rapid tomography but with lower coherence CTF represents higher frequencies well, but reconstruction artefacts dominate the reconstruction of the only partially coherent data.

*Fourier shell correlation and power spectral density calculations.* To measure resolution we performed Fourier Shell Correlation (FSC) analysis (Supplementary Fig. 7) as described before[104,105]. In brief, for each measurement (Supplementary Fig. 7), two paired volume ROIs acquired independently and containing data of the same volume were generated. The ROIs were large enough to contain relevant features (500 voxels wide cubes in all cases except for LXRT, where ROIs were 100 voxels wide) and their contents were checked to ensure they contained biological tissue data throughout. In SXRT and LXRT datasets, the two paired ROIs were generated reconstructing half-tomograms composed by the odd and even projection angles, respectively. A FSC analysis was then performed on the paired ROIs. Resolution readouts were obtained for each FSC curve at the spatial frequency the FSC curve fell below the resolution criterion. Four commonly used resolution criteria were applied to most cases: 1 bit, ½ bit, 3 sigma and 1/7[104,105]. For LXRT, the reconstruction algorithm applied to the half-tomograms was not fully editable and possibly contained some common filtering step, which rendered artifactual high correlations at even 1/(2*pixel) frequency. The average of the FSC curves obtained from three ROI pairs was normalized assuming the correlations at high frequencies [1/(3*pixel) 1/(2*pixel)] would be null. Then, that same normalisation was applied to all individual traces and individual resolution measurements were extracted from the latter.

Power spectral density (PSD) was calculated on flat-field corrected raw projections and phase maps (Supplementary Fig. 13). The x-axis ('spatial frequency') was calculated from a 2D fast FFT and divided by 2000 (~half the size of the FFT image).

**EM imaging**. Three serial block-face electron microscopy (SBEM) datasets were obtained for this study, to explore in detail the olfactory bulb (C525a,b) and the hippocampus (C556) of samples previously imaged with other imaging modalities.

In all cases, the parent stained sample was scanned with LXRT (Zeiss Versa 510) and a trimming plan was designed to render the regions of interest optimally accessible for the SBEM. Additional datasets obtained from other modalities (2p ex vivo datasets of regions containing genetically tagged fluorescent glomeruli) were also warped to the same scene in some cases (C525a,b) to determine the ROI (Supplementary Fig. 8a, b).

Several LXRT datasets taken iteratively at different steps of trimming were warped onto the same scene iteratively using BigWarp[59] (Supplementary Fig. 8 c1-c2, d1-d2), guiding the trimming process and ensuring the ROI was preserved at all times. The trimming process involved the use of tools of increasing precision as required: hand-held single-edge blades, ultramicrotome-guided glass knives and a 4 mm diamond knife (Leica UC7 ultramicrotome, Diatome histo knife). The final sample was a <1 mm³ tissue cuboid with a diamond knife-polished surface displaying some tissue. The sample sides were then covered with a silver colloidal suspension (SPI supplies), and the sample was sputter-coated with 10 nm platinum (Quorum Q150R S).

All SBEM datasets were obtained with single images covering the desired field of view. Datasets C525a and C556 were acquired using a 3View2—variable pressure Zeiss Sigma SBEM using focal charge compensation[30]. In both cases, datasets were acquired using a 2.5 kV electron beam and a 30 μm aperture, and multiple ROIs were acquired at high and low resolution covering narrower and wider fields of view, respectively (Supplementary Table 1). In particular, C525a contained 6004 slices across 11 concatenated datasets in z. Seven high-resolution regions (acquired in z-order 1, 2, 3, 5, 7, 9, 11) were composed altogether by 2078 images acquired at 2.0 μs/pixel, each covering (200 μm)² with (10 nm)² pixels, separated in z by a slice thickness of 40 nm. A reconstructed continuous low-resolution dataset contained 4970 images acquired at 0.5 μs/pixel covering (737 μm)² with (80 nm)² pixels, separated in z by a slice thickness of 80 nm. Acquiring the C525a datasets took 25 days. C556 was composed by one low-resolution dataset and three high-resolution datasets from narrower regions in the same volume, all containing 787 images separated in z by a slice thickness of 50 nm. The images of the low-resolution dataset were acquired at 0.5 μs/pixel and covered (655 μm)² with (80 nm)² pixels. The images of the high-resolution dataset were acquired at 2.0 μs/pixel and covered (82 μm)² with (10 nm)² pixels. Acquiring the C556 datasets took 5 days. C525b was acquired using a 3View2 - Zeiss Merlin SBEM under high vacuum, a 2.5 kV electron beam with a landing current of 0.5 nA and a Gatan OnPoint backscattered electron detector. The dataset was composed by 9899 images acquired at 0.5 μs/pixel covering (700 μm)² with (50 nm)² pixels, resulting in a dose of 0,62e⁻/nm², separated in z by a slice thickness of 50 nm. Acquiring the C525b dataset took 18 days.

Additionally, the micrographs shown in Supplementary Fig. 6a-b were taken from C046, a SBEM dataset previously reported in reference[106].

*Image registration.* Each SBEM dataset was registered using Voxelytics Align (scalable minds, Potsdam, Germany), which uses pairwise feature matching between adjacent slices, RANSAC inlier detection and global relaxation to create mesh-tessellated affine transformations. When a number of slices had to be discarded, the closest non-discarded slices were duplicated in the final dataset thereby keeping the total number of slices unaltered.

C525a datasets (Supplementary Fig. 4a2) were registered by removing 3 slices (Supplementary Fig. 4a4). C556 datasets were registered without any slice loss (Supplementary Fig. 4a5).

For C525b (Supplementary Fig. 4a1), 595 slices (6%) had to be discarded to register the dataset (Supplementary Fig. 4a3), generating one single continuous gap of 10 slices (0.5 μm) and an average gap size of 2.7 slices (135 nm). These slices were detected by looking at maximum variation and at minimum matched features between any two adjacent slices. These artefact detections and the registration were run on a masked version of the dataset only containing tissue voxels. This mask was essential to avoid picking up statistics from imaging artefacts present in areas filled with resin (such as charging artefacts). The final step involved correcting for lateral drift across z, which is most likely a registration artefact arising from the presence of features in the dataset that have a preferred orientation.

*Fourier shell correlation.* Resolution was estimated using FSC as described above except that in the registered SBEM dataset (50 nm voxels), the two ROIs were generated by extracting the odd and even slices and further downscaling by 1/2 in x, y (and therefore the voxel size of the ROIs became 100 nm isotropic).

**Data processing of SXRT and EM data**. Datasets were stored as a chunked file format with a resolution pyramid (wkw) and explored in webKnossos[107].

*Warping.* Linked datasets obtained with different imaging modalities were warped into a common framework using BigWarp[59] (Supplementary Fig. 4b). Conserved landmarks were manually identified, and a first affine transformation of the moving dataset was generated allowing to find further local paired landmarks between both datasets. This iterative process of landmark deposition and dataset transformation was performed until no observable improvement in overlap was

found, using 11–171 paired landmarks. The number of landmarks depended on the modalities being warped: 27–53 landmarks were used for warping 2-photon datasets of fixed tissue into a LXRT (Supplementary Fig. 4b1-b3, b8); 11–25 to warp LXRT and SXRT (b4-b5, b8), 50–51 to warp SXRT and SBEM, and 171 to warp in vivo 2-photon to SBEM (b6-b7, b9). The nature of landmarks also depended on the modalities being warped, since this relies on conserved features across both modalities: glomerular contours and tissue boundaries were features present in all datasets, blood vessels were well defined in the in vivo 2-photon dataset, and these and nuclei were easily found in SXRT and SBEM. This set of landmarks was then used to generate a warp field to move between both datasets. The Bigwarp engine was integrated into Matlab using custom code and integrated with a webKnossos skeleton analysis toolbox[6]. We constructed a graph with nodes representing datasets and edges representing transform functions, either involving spline transformations as introduced above or scale-offset transformations. This graph contained all information required to transfer annotations from the coordinate system of a dataset to the coordinate system of an arbitrary linked dataset. This allowed warping skeletons between any connected pair of datasets, a key tool for the comparative analyses presented. In some cases, the image data from regions of interest spanning part or the whole moving dataset were warped into the target dataset's space and rendered in a common viewport using Amira (Thermo Fisher) for illustrative purposes.

### Data analysis of SXRT and EM data

*Apical dendrite tracing and SXRT traceable length analyses.* Mitral cell apical dendrites were traced in SBEM and SXRT datasets. All datasets were processed and managed using a webKnossos data storage, display and annotation infrastructure[107]. Somata of mitral cells were identified based on their histological and morphological features[108], mainly a large cell body located in a monolayer parallel to the glomerular layer, 200–400 µm below it, with a large, pale nucleus containing a prominent nucleolus and surrounded by abundant cytoplasm. All mitral cell somata (372) in the SBEM dataset (C525b) were manually tagged by creating a node in their nucleus and their names and coordinates were stored in a webKnossos skeleton file (*.nml). 50 somata were randomly selected without replacement for this analysis and saved as a new skeleton. This file was warped onto the SXRT dataset space using the integrated warping toolbox. This enabled manually tracing the same cell in the native version of each imaging modality. Three trained tracers were assigned the task of manually tracing all apical dendrites in both SBEM and SXRT datasets. Briefly, it involved identifying the apical dendrite that arises from the cell body of the seeded nucleus, and then tracing that apical dendrite centrifugally until reaching the glomerulus without creating any branches. Once inside the glomerulus, the most prominent dendrite was to be traced as well. All SBEM skeletons (with 3 traces per neuron) were pooled and manually curated (all could be reconstructed: 34/50 were correctly traced by all, 16/50 were correctly traced by two tracers). The longest tree of the correctly traced ones was proofread and chosen as consensus apical dendrite for that one neuron, generating ground truth apical dendrite skeletons for all 50 mitral cells. These ground truth skeletons were warped into the SXRT space using the integrated warping toolbox described above. For each tracer, the apical dendrites traced in the SXRT dataset were paired with the warped ground truth (SBEM) apical dendrite of the same cell. Both trees would start at the same position (nucleus).

A polynomial cubic spline function (Matlab File Exchange) was fitted to both trees, and evenly spaced nodes were created every 320 nm. For every node in the SXRT-traced tree, the euclidean distance to the closest node in the warped EM tree was calculated and plotted as a function of the position in the SXRT-traced tree. These distances displayed a bimodal distribution with the two peaks arising from correctly and incorrectly traced dendrites respectively. A tree was considered "lost" when the closest distance to the consensus tree was >12 µm, and the distance from the soma to the node at which it was lost was defined as the correctly traced distance. Most tree losses took place within the first 40 µm of dendrite, which can be attributed to the beginning of the apical dendrite. We named this challenge as the "linkage" between the soma and the apical dendrite. We called "linked" all trees that were correctly traced for at least 35 µm. This enabled splitting all trees between kept or lost, linked or not linked, or shorter than both thresholds 'linked' and 'kept' (Supplementary Fig. 3a).

A similar approach was used in the hippocampus (Supplementary Fig. 3c–h). In this experiment, the EM dataset was configured so the *xy* plane matched the direction of CA1 apical dendrites and the *z*-axis evolved almost perpendicular to the pyramidal layer (Supplementary Fig. 3c, d). A bounding box of $180 \times 200$ µm² in *x, y* and spanning the entire *z* of the dataset (39 µm) was defined so it would contain the CA1a region. Pyramidal neurons were easily identifiable based on their large soma with a nucleolus and condensed chromatin inclusions located in a packed layer with a clearly distinguishable apical dendrite arising from the cell body. All 175 CA1 pyramidal somata in the bounding box were tagged by creating a node in their nucleus. One trained tracer was assigned to trace all their dendrites in both EM and SXRT datasets (Supplementary Fig. 3d). In EM, the dendrites of 132 somata could be traced, generating ground truth apical dendrite skeletons for these 132 CA1 pyramidal neurons. 90 of these could be traced in the SXRT data. In this case, a warped version of the SXRT dataset on the EM space was also used to complement the tracings when helpful. The distances between SXRT and EM trees

also displayed a bimodal distribution, which was used to set the "lost" threshold to 6 µm. The "linkage" threshold was set to 35 µm (Supplementary Fig. 3e).

The same tracing and analysis methods were used to compare the tracing accuracy for apical dendrites of OB mitral cells in SXRT datasets acquired in two different microtomography beamlines (Supplementary Fig. 3i–l) (as in previous OB analyses: interpolation step = 320 nm, lost threshold = 12 µm). Here, I13-2 tracings were used as references and TOMCAT tracings were evaluated against them. Since the distance analysis only applies to dendrite tracing pairs, only cells traced in both datasets were analysed (33 out of 50 randomly seeded mitral cell somata). From these, 91% of the dendrites were accurately traced in both SXRT datasets (30 out of 33) (Supplementary Fig. 3l).

### Correlative SXRT-SBEM analyses of hippocampal CA1 neurons

*Soma depth of CA1 pyramidal neurons.* A linear regression line was fitted to the x,y coordinates of all seeded CA1 pyramidal neuron somata. Soma depth was measured as the euclidean distance between a soma and the regression line. Here, the soma z coordinate was ignored as apical dendrites were approximately parallel to the xy image plane. Deeper cells have more positive scores. A subset of the 90 CA1 apical dendrites traced in the SXRT dataset passed through the high-resolution SBEM dataset. Among them, seven dendrites with varying soma depths were chosen for further spine and spine apparatus analyses.

*Dendritic spine tracing and spine apparatus tagging.* By warping SXRT apical dendrite tracings to the high-resolution SBEM dataset space, dendrites corresponding to the seven target cells were identified. Like in apical dendrite tracings, one trained tracer manually annotated all dendrite branches, dendritic spines and spine apparati on webKnossos. For each of the seven dendrites, first the dendritic shaft was delineated by a linear trace. Second, spines were annotated by placing a node in the spine head, connected to a node in the spine neck, connected to the dendritic shaft through a perpendicular intersection. Spines containing more than one spine apparatus cistern inside its head or neck were annotated as having spine apparatus.

*Density of spines and spine apparati.* Dendrite skeletons were split into individual branchlets and grouped by dendrite type (trunk or apical oblique). For each dendrite branchlet, the number of spines and spines with spine apparatus were counted. Branched spines with spine apparatus in both of its heads were counted as one spine with spine apparatus. Both inter-spine and inter-spine apparatus distances were measured. In total, from the seven cells, 75 dendrite branchlets, which were 2 mm in length in total, were inspected and their 3019 spines and 417 spine apparati analysed for density and distribution.

Spine density was defined as the total number of spines divided by the total shaft length pooled across all dendrite branchlets of the same order for each cell.

Spine apparatus density was similarly defined as the total number of spines possessing a spine apparatus divided by the total length of the dendritic shaft pooled.

The maximum inter-spine distance was calculated for each dendrite branchlet and plotted against the branchlet length (Supplementary Fig. 9e1-2). In total, 53 out of 58 apical oblique dendrites contained more than one spine (i.e. at least one inter-spine distance value) and were included in the analysis.

*Distribution of spines and spine apparati.* If spines emerged from the shaft randomly, according to a homogeneous Poisson process, inter-spine distances would be distributed according to the exponential distribution $f(x) = \lambda e^{-\lambda x}$, where $\lambda$ is the average inter-spine distance. For each cell, the *p*-value from one sample Kolmogorov–Smirnov test between inter-spine distance distribution and a fitted exponential distribution was calculated (Supplementary Fig. 9f1).

Similarly, having a spine apparatus may also occur randomly in spines. To assess that, a similar *p*-value was retrieved for each cell by comparing the inter-spine apparatus distance distribution to the same fitted exponential distribution as above (Supplementary Fig. 9f2). All density and distribution characteristics were analysed in light of soma depths.

**Reporting summary.** Further information on research design is available in the Nature Research Reporting Summary linked to this article.

## Data availability

Source data of the graphs presented in the main figures are provided as a Source Data file. The datasets and major annotations reported in this study are accessible through the associated code repository (see Code Availability).

## Code availability

Warping landmarks and code is accessible for public use[109], with the latest version available at https://github.com/FrancisCrickInstitute/warpAnnotations.

The following data acquisition and analysis software was used:

2-photon data was acquired with commercially available Scientifica software (SciScan 1.3).

Epifluorescence data was acquired using the commercially available ZEISS ZEN software.

LXRT tomograms were acquired in a ZEISS Versa 510 micro-CT using ZEISS Scout and Scan and reconstructed using ZEISS Reconstructor.

SXRT tomograms were acquired using GDA (at Diamond, http://www.opengda.org/) and reconstructed with savu (at Diamond I13-2, https://github.com/DiamondLightSource/Savu) and GridRec (at PSI TOMCAT, https://github.com/esther279/Gridrec-MS). At ID19 data was collected using Bliss (https://bliss.gitlab-pages.esrf.fr/bliss/master/) and reconstructed with Nabu (https://tomotools.gitlab-pages.esrf.fr/nabu/). At ID16A data was acquired using Spec and Tango (https://www.tango-controls.org/), and reconstructed using Octave and PyHST (https://software.pan-data.eu/software/74/pyhst2). Reconstructed tomograms of the same specimen were stitched using an implementation of NRStitcher (https://github.com/arttumiettinen/pi2).

SBEM datasets were acquired on a variable pressure Sigma SEM 3View2 using SBEMimage software (https://github.com/SBEMimage) and on a high vacuum Merlin SEM 3View2 using ZEISS SmartSEM (version 5.07) and Gatan DigitalMicrograph (version 3.32.2403.0).

All datasets were converted into a pyramidal file format using Voxelytics and explored and annotated through webKnossos (https://github.com/scalableminds/webknossos).

Data analysis was performed using Python (version 3.8.3), Fiji (ImageJ 1.53c), suite2p, MATLAB (2019a) and Amira (6.7.0). The toolbox for warping skeletons as described in "Methods" section "Warping" is based on BigWarp (https://github.com/saalfeldlab/bigwarp) and its MATLAB implementation derived from https://gitlab.mpcdf.mpg.de/connectomics/L4dense. This implementation is accessible for public use alongside the warping landmarks, datasets and annotations described in this study.

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

## Acknowledgements

The authors are grateful to the advanced light microscopy, biological research, electron microscopy and scientific computing science technology platforms of the Francis Crick Institute. We thank Ede Rancz, John Bogovic, Kevin Briggman and Sina Tootoonian for discussion, Marta Majkut and Alexander Rack for support and imaging experiments at ID19, and Peter Cloetens and Pierre Paleo for discussion and support with data analysis and synchrotron imaging. This research was funded in whole, or in part, by the Wellcome Trust (FC001153 and 110174/Z/15/Z to A.T.S.; FC001999 to L.M.C.; 090843/F/09/Z to T.W.M.). For the purpose of Open Access, the author has applied a CC BY public copyright licence to any Author Accepted Manuscript version arising from this

submission. This work was carried out with the support of Diamond Light Source, beamline I13-2 (proposal 20274), the TOMCAT beamline of the Swiss Light Source at the Paul Scherrer Institut (proposal 20190417) and the ID16A and ID19 beamlines of the European Synchrotron Radiation Facility (proposal ih-ls-3253). This work was supported by the Francis Crick Institute, which receives its core funding from Cancer Research UK (FC001153 to A.T.S.; FC001999 to L.M.C.), the UK Medical Research Council (FC001153 to A.T.S., FC001999 to L.M.C.), and the Wellcome Trust (FC001153 to A.T.S., FC001999 to L.M.C.). It was also supported by the UK Medical Research Council (MC_UP_1202/5 to A.T.S.), by the Gatsby Charitable Foundation (GAT3361 to T.W.M.) and by a DFG postdoctoral fellowship to T.A. A.P. acknowledges funding from the European Research Council under the European Union's Horizon 2020 Research and Innovation Programme (852455).

## Author contributions

C.B. and A.T.S. conceived and designed the research; C.B., T.A., A.P., C.J.P., M.-C.Z., M.S., A.B. and A.T.S. performed experiments and acquired data; C.B., T.A., A.P., Y.Z., M.B., N.R., M.-C.Z., I.W., M.S., A.B. and A.T.S. analysed and/or interpreted data; M.B. and N.Z. developed new software tools; C.B. and A.T.S. drafted the manuscript; C.B., T.A., Y.Z. and M.S. made figures; A.P., M.S., C.R., T.M., L.C. and A.T.S. provided supervision and advice and all authors (C.B., T.A., A.P., Y.Z., C.J.P., M.B., N.R., M.-C.Z., I.W., M.S., A.B., C.R., T.M., L.C. and A.T.S.) reviewed and provided input to the manuscript.

## Funding

## Competing interests

The authors declare no competing interests.
