## [Peer Review File · Nature Communications]

REVIEWER COMMENTS

Reviewer #1 (Remarks to the Author):

The main focus of the manuscript is the correlative brain-imaging between synchrotron tomography, electron microscopy and in-vivo 2-photon Ca imaging.

Although the paper is interesting, it is neither original nor good enough for publication in Nature Communication.

General considerations:

The correlation between synchrotron tomography, in particular synchrotron phase contrast tomography, and electron microscopy has already been published by some of the authors of the same paper, but also by others (not even cited in this paper). Therefore this first part, which unnecessarily absorbs the main part of the Results section (6 figures over 7 total figures), is neither innovative nor original. The last part, potentially more original, is not enough described (see technical considerations) and the specific application is not significant for a general use of the approach.

The manuscript, including the title, is not clearly written, neither for an expert of the field nor for a wide-public reader. Mix-up between X-ray tomography and X-ray Phase Contrast Tomography confuses the reader.

Technical considerations:

How many animals and samples were used for in-vivo and ex-vivo experiments? Is the result statistically significant if it is only referred to one sample? It is not clear whether the olfactory bulb and the other parts of the brain that were investigated come from the same animal.

The authors used synchrotron X-ray computed tomography (SXRT) of objects stained with heavy metals. In the "Method" section and subsection "SXRT imaging" the authors describe phase tomographic reconstruction with the Paganin method. This let understand that X-ray Phase Contrast is used, although it was not clear from the beginning. However, the Paganin method is not ideal in strong-absorbing cases.

The introduction is just a list of the main characteristics of the different microscopic techniques typically used to image the nervous system. It does not clearly describe the peculiar features of the multimodal approach that should be the focus of this work.

The most critical technical issue to take care of in the proposed approach is the research of the same sample areas analysed with the different techniques (2 different synchrotrons for SXRT, EM, in-vivo 2-photon imaging). This critical point deserves more attention and wider description.

Owing to all the above reasons this manuscript is not suitable for publication In Nature Communications

Reviewer #2 (Remarks to the Author):

Using mouse neural circuits as a model system, the authors provide a comprehensive pipeline for the correlation of in vivo functional 2-photon microscopy, large field of view synchrotron X-Ray tomography (SXRT) and high resolution, serial block-face electron microscopy (SBEM). Correlating mm³ scale SXRT images with nm³ scale SBEM, the authors show, fast, reliable tracing of apical dendrites over 100's of μm of damage free tissue. The manuscript gives biological context by investigating pyramidal neuron depth, with superficial cell bodies have greater density of spines with spine apparatus, than deeper ones. Additionally, the authors perform elegant in vivo 2-photon microscopy of the olfactory bulb in genetically labelled mice, addressing a key challenge of relating, sometimes ambiguous, functional imaging data to sub-cellular features.

Although an established technique, synchrotron X-ray imaging is a hot topic for imaging soft tissue right now. The authors have included recent publications, using various X-ray imaging techniques and approaches, for the reader to refer to. All approaches involve many challenges and limitations, also discussed.

The manuscript is a significant undertaking, showcasing complex techniques in Bio-Imaging, data processing and data analysis across multi-scale modalities. The clarity of presentation, in depth methodology and extensive supplementary material of the manuscript will provide the reader with enough information to assess if these techniques are applicable to their studies. Each technique in isolation requires specialist technology and technologists; the manuscript is a reflection of a broad collaborative approach, which is commendable and effective.

The novelty of the manuscript is in its reliability to bridge the gap between functional in vivo imaging at the mm³ scale and high resolution nm³ scale volume EM, with non-destructive, fast, x-ray imaging (13-20mins per data set), making it of interest not only to the neuroscience community, but anyone imaging soft tissue. In my opinion the manuscript should be published without further review. I have suggested some minor points for consideration.

Fig2, d2 – arrowhead to guide eye to apical dendrites discussed in text.

Fig5, b - Really busy, perhaps crop/shrink image a, as this has already been shown previously, to give more room for b. as arrowheads on histogram not easy to see.

5d fix a. o. typo.

Alison June Beckett

Reviewer #3 (Remarks to the Author):

The authors present an impressive and innovative approach to multi-scale and multi-modal imaging. The relevance and interpretation for neuroscience is convincing.

The fact that the authors can bridge live imaging of the olfactory bulb to high resolution and volume EM, and show the

advantage of XRT large volume overviews in choosing the regions of interest is very convincing and can pave the way for important future work. Image data is evaluated with high-end quantitative analysis and valuable details and protocols are described and disclosed, but some important information is missing, and not all of the quantifications are adequately described /presented (see below). In view of the tremendous amount of data and work compiled, it is not surprising that the organisation of material and figures presents some challenges.

I would like to recommend the work for publication, but can do so only subject to a major revision targeting two major points /deficits:

1.) Completeness of XRT description

Given the fact that the central claim of novelty is SXRT, and the correlative workflow, the SXRT is insufficiently described/quantified.

Essential information such as the resolution achieved is missing:

-Fourier Shell Correlation and power spectral density should be used to estimate resolution for LXRT and SXRT (both beamlines); resolution is visibly much, much worse than the pixel, putting in question the choice of the objectives used

-For the heavy metalized samples phase contrast effects are not obvious, and should be illustrated by showing projections, and a reconstructed projection,

-The assumption of a homogeneous object is probably not given. Please comment.

- The assumption of direct contrast regime ($F > 1$) is not given for the chosen pixel size. Please give Fresnel numbers. Did you try different phase retrieval approaches, such as CTF valid also for lower F ?

-estimate delta and beta for different labeled components based on what is known for uranyl and osmium stoichiometry and concentration

-give measured transmission for the samples & attenuation coefficients

-include a simulation of which contrast you would expect as a function of E (for the modeled optical tissue) so that choice of photon energy can be rationalized

- give details of the tomographic reconstruction including post-processing (such as ring filter)

2.) Quality of the Presentation

The lack of detail and precision for the SXRT/LXRT is contrasted by information overflow in other parts of the MS /SM.

Generally I find that the data and method presentation is too packed, and not organized well enough. Think about using more tables, less crowded composite graphics; larger Figures, where essential (e.g. SXRT/LXRT slices); all essential graphical displays should be larger. In case of space constraints, leave

something less important out.

Supp.Fig.9, for example: As such, a1 and a2 are just not informative. Take one curve, explain it well and in detail, including acronyms of the legend, and then

show the differences between different cells in from of a table.

I find the entire description of the inter-spine distribution is quite incomprehensible.

As a test whether the presentation/description is fit for publication: ask each other within the co-author team, whether everybody can understand each other's part well.

Fig.5, for example: (i) I cannot find the description, p-values refer to which test? what is the finding exactly ? What do we learn, by plotting SA density as a function of some depth?

Any hypothesis or interpretation associated with this?

(g,j) The segmentations of the high res EM are not explained, what are we seeing. Of course, once you know how a spine apparatus looks in EM, you can guess; but overall

please carefully choose, describe, and if in doubt leave out. The high res EM of the spines is nice, why not present it larger at the expense of something else.

Sometimes less is more.

Reply to Reviewers

We want to thank the editor and all three reviewers. We have addressed all comments with new experiments both from the beamlines I13-2 and TOMCAT as well as from new beamlines ID19 and ID16A at ESRF, new analysis, simulations as well as significant improvements to the text and figure organisation. The new figures are summarised below for convenience in the table **Figure_R0**. We are hopeful that the substantially revised manuscript will be suitable for consideration for Nature Communications.

Rebuttal Figure	Title	Manuscript
Figure R_1.1	Sister mitral cells from genetically-targeted glomeruli.	Supp. F5
Figure R_1.2	Sample Table.	Supp. T1
Figure R_1.3	Transmission of heavy metal-stained brain samples.	text, p13-14
Figure R_1.4	Optimisation of acquisition and reconstruction parameters for SXRT datasets.	Supp. F15
Figure R_1.5	Comparison of tomography reconstruction algorithms.	Supp. F12
Figure R_1.6	Comparison of correlative multimodal imaging (CMI) pipelines.	Supp. F1
Figure R_3.1	Resolution of LXRT, SXRT and SBEM datasets.	Supp. F7
Figure R_3.2	Power Spectral Density.	summarised in Supp. F13a1-3
Figure R_3.3	SXRT at 20x and 40x.	Supp. F14
Figure R_3.4	Sample homogeneity.	Supp. F16
Figure R_3.5	Experimental parameters at different beamlines.	Supp. T2
Figure R_3.6	Comparison of reconstructions with CTF and Paganin phase-retrieval.	Supp. F13
Figure R_3.7	Chemical compounds used in the staining, dehydration and embedding protocol.	text, p9
Figure R_3.8	Estimated average composition of the specimen.	text, p9
Figure R_3.9	Measured and estimated transmission.	text, p9
Figure R_3.10	Density of hard epon resin.	text, p12
Figure R_3.11	Simulation of X-ray contrast.	text, p12
Figure R_A1	ROIs extracted from an individual tile of two specimens imaged at DIAMOND I13-2.	summarised in Supp. F7a
Figure R_A2	ROIs extracted from an individual tile of an specimen (C432) imaged at TOMCAT.	summarised in Supp. F7a
Figure R_A3	ROIs extracted from the registered C525b dataset imaged with SBEM.	summarised in Supp. F7b
Figure R_A4	ROIs extracted from a specimen (C435) imaged with LXRT.	summarised in Supp. F7c

Figure_R0: Figures addressing reviewer's comments.

REVIEWER COMMENTS

Reviewer #1 (Remarks to the Author):

The main focus of the manuscript is the correlative brain-imaging between synchrotron tomography, electron microscopy and in-vivo 2-photon Ca imaging. Although the paper is interesting, it is neither original nor good enough for publication in Nature Communication.

General considerations:

The correlation between synchrotron tomography, in particular synchrotron phase contrast tomography, and electron microscopy has already been published by some of the authors of the same paper, but also by others (not even cited in this paper). Therefore this first part, which unnecessarily absorbs the main part of the Results section (6 figures over 7 total figures), is neither innovative nor original. The last part, potentially more original, is not enough described (see technical considerations) and the specific application is not significant for a general use of the approach.

Thank you for pointing this out. We have substantially expanded the description of the correlative in vivo physiology-X-ray aspect of the paper. We have also taken the comments on board and streamlined our description as well as doubled down on our efforts to include the relevant literature in the introduction.

The manuscript, including the title, is not clearly written, neither for an expert of the field nor for a wide-public reader. Mix-up between X-ray tomography and X-ray Phase Contrast Tomography confuses the reader.

Thank you for pointing out this source of confusion. We have clarified our language and now consistently use SXRT for the propagation-based phase contrast X-ray tomography performed at the parallel beam micro-tomography beamlines I13-2, TOMCAT and ID19. We have also provided substantially more detail about the phase contrast methods and phase retrieval algorithms employed (and their limits).

Technical considerations:

How many animals and samples were used for in-vivo and ex-vivo experiments? Is the result statistically significant if it is only referred to one sample? It is not clear whether the olfactory bulb and the other parts of the brain that were investigated come from the same animal.

We have now very clearly indicated which samples stem from which animal, including an overview table with all samples (including newly acquired and analysed ones) and the detailed figure references as new **Supplementary Table 1** in the revised manuscript. Indeed, the approach is very reproducible and we appreciate that this wasn't clear from the original manuscript. We have now performed more SXRT experiments and traced apical dendrites from sister cells (cells projecting to the same glomerulus) in 10 samples from 9 animals to illustrate reproducibility in the new **Supplementary Figure 5**, shown below as **Figure R_1.1**. We have also included the full sample table (**Supplementary Table 1** in the revised manuscript) below as **Figure R_1.2**. We hope that this makes clear that we describe a robust and reproducible approach.

Figure R_1.1: Sister mitral cells from genetically-targeted glomeruli.

(a) SXRT virtual coronal cross-sections of 10 olfactory bulb samples from 9 mice (in one case, both brain hemispheres were sampled). In all cases the genetically-identified glomerulus (MOR174/9 or M72) could be identified through a correlative *ex vivo* 2p, LXRT and SXRT approach. Two sister mitral cells (blue, orange) were traced on a glomerulus in each dataset, and a grey mesh indicates the contour of that glomerulus traced in the SXRT dataset. In those cases where the glomerulus shown corresponds to the genetically-targeted one, coloured meshes display the glomerular contour of the fluorescently-labelled glomerulus, traced on the *ex vivo* 2p dataset and warped onto the SXRT space (green for MOR174/9, yellow for M72).

(b) Close-up view of the same annotations, more clearly displaying the cell bodies, apical dendrites and glomeruli.

D, dorsal; *V*, ventral; *L*, lateral; *M*, medial.

Specimen	animal_ID	age (w)	gender	hemisphere	location	figures
C435	ASAG21.3a	12	male	left	first dorsal slice, olf. bulb	SuppF7, SuppF13, SuppF14, SuppF15
C417	ASAH16.9a	8.3	male	right	first dorsal slice, olf. bulb	SuppF12, SuppF13
C406	ASAH33.1a	9.3	male	left	first dorsal slice, olf. bulb	SuppF5
C410	ASAH33.1b	9.3	male	left	first dorsal slice, olf. bulb	SuppF5
C319	ASAK1.2h	13.2	male	left	first dorsal slice, olf. bulb	SuppF16
C458	ASAM11.3c	9	male	left	first dorsal slice, olf. bulb	SuppF5
C525	ASAM15.2a	10	male	left	first dorsal slice, olf. bulb	1, 2, 3, 6, 7, SuppF1, SuppF2, SuppF3, SuppF4, SuppF5, SuppF6, SuppF7, SuppF8, SuppF10, SuppF11
C414	ASAM3.2a	10	male	left	first dorsal slice, olf. bulb	SuppF5
C432	ASAM3.2g	11	female	left	first dorsal slice, olf. bulb	SuppF3, SuppF5, SuppF7
C433	ASAM3.2g	11	female	right	first dorsal slice, olf. bulb	SuppF5
C488	ASAM3.9d	8	male	left	first dorsal slice, olf. bulb	SuppF5
C450	ASAM5.4d	12	female	left	first dorsal slice, olf. bulb	SuppF5
C420	ASAM7.3i	12	female	left	first dorsal slice, olf. bulb	SuppF5
C543	ASAU7.2a	10	male	n/a	coronal slice, cortex and striatum	5, SuppF1
C555	ASAU7.2c	10	male	n/a	coronal slice, cortex and anterior hippocampus	5, SuppF1
C556	ASAU7.2c	10	male	n/a	coronal slice, cortex and medial hippocampus	4, SuppF1, SuppF3, SuppF4, SuppF9
C557	ASAU7.2c	10	male	n/a	coronal slice, cerebellum	5, SuppF1
C046	BRAC	12	male	n/a	coronal slice, olfactory bulb	SuppF6

Figure R_1.2: Sample Table

Details of all specimens reported in the study. Specimens are ordered by animal identity, and all specimens belonging to the same animal display the same background on the first column.

The authors used synchrotron X-ray computed tomography (SXRT) of objects stained with heavy metals. In the “Method” section and subsection “SXRT imaging” the authors describe phase tomographic reconstruction with the Paganin method. This let understand that X-ray Phase Contrast is used, although it was not clear from the beginning. However, the Paganin method is not ideal in strong-absorbing cases.

Thank you for highlighting this important aspect. We now explicitly report absorption for our samples (**Figure R_1.3** and **page 13** of the revised manuscript). To test the potential contribution of phase contrast, at l13 we obtained reconstructions at different propagation

distances (**Figure R_1.4, a**). Small distances (~absorption regime) do not allow us to resolve details to the same extent as longer propagation distances where the partial coherence of the beam allows for the use of phase contrast reconstruction (**Figure R_1.4 b,c**). Thus, despite the (relatively) strong absorption, the Paganin method allows us to recover improved images compared to the pure absorption regime. In order to more quantitatively assess the regime we were in, we now provide Fresnel numbers for all beamlines and recording configurations employed (included as part of the new experimental parameters **Supplementary Table 2** in the revised manuscript). We now explicitly describe the conditions, assumptions and limitations on **pages 13-14** of the revised manuscript and with the new **Supplementary Figures 12-16**. Please see also the detailed response to the comments of Reviewer 3 below.

Figure R_1.3: Transmission of heavy metal-stained brain samples.

Delta/beta values for Paganin reconstruction were estimated from the sample composition (see below). As these estimates are difficult to make, we varied the delta/beta ratio substantially to empirically obtain the best reconstruction quality (**Figure R_1.4**). We now describe this explicitly in the revised manuscript on **pages 13-14**.

Other phase reconstruction approaches might allow further improvement of the final image quality. We thus now also employed CTF for phase retrieval. While CTF (as also discussed in the response to Reviewer 3) has the potential to reconstruct higher resolution features, Paganin reconstruction is more robust against lack of temporal coherence / non-monochromatic beam. To compare CTF with Paganin we initially performed experiments in a situation with further improved beam coherence, at ID16A (1% $\Delta E/E$). Indeed, under these circumstances CTF improves reconstruction compared to Paganin (**Figure R_1.5**). However, for many of the widely accessible μ CT beamlines, monochromators are often not employed (as in our experiments at I13). This can of course be due to a number of different reasons including experimental simplicity / robustness. Moreover, in the absence of a monochromator, flux is substantially higher allowing for rapid tomography, however with reduced temporal coherence. Under these circumstances we believe that Paganin allows for a more robust reconstruction albeit with somewhat reduced fine features compared to CTF under ideal conditions. We now highlight these important points regarding the suitability of different phase retrieval approaches for heavy metal-stained tissue in the discussion of the revised manuscript. We also include the new data and new analysis figures as new **Supplementary Figures 12 and 15**.

Figure R_1.4. Optimisation of acquisition and reconstruction parameters for SXRT datasets.

(a-c) Different sample-detector distances (a) were tested at the different beamlines (cases shown relate to tests performed at I13-2). Using the same sample, a tomogram was acquired at each distance, reconstructed, and the same region was located in all reconstructions. The histological patterns revealed across ~250 μm landscapes (b) enabled judging quality variations across extreme parameter values, but examining details at the 20 μm scale such as cell nuclei (c) was necessary to decide on the optimal sample-detector distance (highlighted). (d-f) Different delta/beta ratios (d) were tested empirically at the

different beamlines (cases shown relate to tests performed at I13-2). A raw tomogram (acquired at the optimal sample-detector distance), was reconstructed several times using the different delta/beta ratios and the same region was located in all reconstructions. The histological patterns revealed across $\sim 250 \mu\text{m}$ landscapes (**e**) enabled judging quality variations across extreme parameter values, but examining details at the $20 \mu\text{m}$ scale such as cell nuclei (**f**) was necessary to decide on the optimal delta/beta ratio in the implemented reconstruction algorithm (highlighted).

(**g**) Raw projection (**g1**) and corresponding PSD (**g2**) and azimuthally-averaged PSD (**g3**) for a sample-detector distance of 14 mm at Diamond I13-2. (**g4-g6**) same as (**g1-g3**) but for a sample-detector distance of 100 mm. Note the gain in information for larger propagation distance.

onl, olfactory nerve layer; *gl*, glomerular layer; *epi*, external plexiform layer; *mcl*, mitral cell layer; *ipl*, inner plexiform layer; *gcl*, granule cell layer.

Figure R_1.5 - Comparison of tomography reconstruction algorithms.

A specimen containing mouse olfactory bulb external plexiform layer was imaged at the nano-imaging beamline ID16A (ESRF) and reconstructed with two algorithms: CTF (**a**) and Paganin (**b**). The same region was located in both reconstructions. The reconstructed field of view is shown in the top panels, and close-up details from each reconstruction method are shown in the bottom panels, respectively.

The introduction is just a list of the main characteristics of the different microscopic techniques typically used to image the nervous system. It does not clearly describe the peculiar features of the multimodal approach that should be the focus of this work.

We have now focussed our introduction as suggested (whilst providing what we hope to be an adequate coverage of the relevant literature).

The most critical technical issue to take care of in the proposed approach is the research of the same sample areas analysed with the different techniques (2 different synchrotrons for SXRT, EM, in-vivo 2-photon imaging). This critical point deserves more attention and wider description.

Thank you for pointing out this indeed important aspect of our work.

We have now included a description of the different imaging modalities (in graphic form) as a new panel in **Figure 1 (Fig. 1d)** and have included additional 2-photon-SXRT-SBEM correlative data in **Fig. 7** of the revised manuscript. Besides, all imaging modalities and their combination is graphically described in **Supplementary Figure 1**, reproduced here as **Figure R_1.6**.

We have extended our coverage of the rationale of correlative multimodal imaging pipelines in the introduction and described in more detail the implications of our findings in the in-vivo 2-photon / SXRT / SBEM experiment in the results and discussion sections.

Finally, we have collected all datasets of the correlative multimodal experiments reported and made them available, alongside the warping code and annotations, in an organised repository:

<https://github.com/FrancisCrickInstitute/warpAnnotations>

Fig. R_1.6 - Comparison of correlative multimodal imaging (CMI) pipelines.

(a) Flowchart diagram (a1), applied to a CMI pipeline including 6 imaging techniques (a2) and to a CMI pipeline including 3 imaging techniques (a3). This diagram shows the elements

that might affect the throughput of a CMI pipeline and is helpful for identifying bottlenecks. Samples are processed before every imaging technique to match compatibility requirements and enhance signal detection. A quality control step is present before any sample preparation and imaging step, aimed to maximise the success rate of the following steps. Raw data has to be processed to generate a curated dataset. Curated datasets can be correlated, thereby enabling to link the information obtained in their respective analyses and augmenting the knowledge extracted from the pipeline.

(b) Correlated datasets in a CMI experiment including 7 modalities **(b1)**, 4 modalities **(b2)**, and in four experiments including the same 2 modalities each **(b3)**. Note that some imaging techniques can image the same specimen at different sampling rates, providing more than one modality (e.g. SBEM high vs low-resolution). Furthermore, some techniques allow imaging multiple regions of interest, thereby providing more than one dataset per modality (e.g. SBEM high-resolution datasets in **b1**). For each dataset, the x axis shows the “voxel size” being the cubic root of the product of the acquired voxel size in x,y,z - thereby representing the length of the side of a voxel if voxels were isotropic. The y axis shows total volume sampled. In this diagram, datasets containing the same number of voxels are distributed along a diagonal. For reference, diagonals hosting datasets of each order of magnitude are shown in shaded greys and their sizes are indicated assuming they are uncompressed 8-bit images. The dataset marker represents the imaging modality: *2p_iv*, 2-photon *in vivo* Ca²⁺; *epi*, epifluorescence of the dissected slab; *2p_ev*, 2-photon *ex vivo* imaging of the fixed slab; *LXRT*, laboratory X-ray μ CTs; *SXRT*, synchrotron X-ray computed tomography with propagation-based phase contrast; *sbemLR* and *sbemHR*, serial block-face electron microscopy at low and high resolution respectively. Datasets spatially correlated are linked with an edge. All plots represent CMI pipelines reported in this study. **(b1)** and **(b2)** contain one single experiment each. In **(b3)**, datasets belonging to each experiment are shown in the same colour.

Reviewer #2 (Remarks to the Author):

Using mouse neural circuits as a model system, the authors provide a comprehensive pipeline for the correlation of in vivo functional 2-photon microscopy, large field of view synchrotron X-Ray tomography (SXRT) and high resolution, serial block-face electron microscopy (SBEM). Correlating mm³ scale SXRT images with nm³ scale SBEM, the authors show, fast, reliable tracing of apical dendrites over 100's of μm of damage free tissue. The manuscript gives biological context by investigating pyramidal neuron depth, with superficial cell bodies have greater density of spines with spine apparatus, than deeper ones. Additionally, the authors perform elegant in vivo 2-photon microscopy of the olfactory bulb in genetically labelled mice, addressing a key challenge of relating, sometimes ambiguous, functional imaging data to sub-cellular features.

Although an established technique, synchrotron X-ray imaging is a hot topic for imaging soft tissue right now. The authors have included recent publications, using various X-ray imaging techniques and approaches, for the reader to refer to. All approaches involve many challenges and limitations, also discussed.

The manuscript is a significant undertaking, showcasing complex techniques in Bio-Imaging, data processing and data analysis across multi-scale modalities. The clarity of presentation, in depth methodology and extensive supplementary material of the manuscript will provide the reader with enough information to assess if these techniques are applicable to their studies. Each technique in isolation requires specialist technology and technologists; the manuscript is a reflection of a broad collaborative approach, which is commendable and effective.

The novelty of the manuscript is in its reliability to bridge the gap between functional in vivo imaging at the mm³ scale and high resolution nm³ scale volume EM, with non-destructive, fast, x-ray imaging (13-20mins per data set), making it of interest not only to the neuroscience community, but anyone imaging soft tissue. In my opinion the manuscript should be published without further review. I have suggested some minor points for consideration.

Thank you very much for your support and encouragement!

Fig2, d2 – arrowhead to guide eye to apical dendrites discussed in text.

We have added arrows to the revised figure pointing to the apical dendrites (now part of **Figure 1**).

Fig5, b - Really busy, perhaps crop/shrink image a, as this has already been shown previously, to give more room for b. as arrowheads on histogram not easy to see.

Thank you. We have substantially simplified the figure (now **Figure 4** in the revised manuscript) and also revised the corresponding **Supplementary Figure 9**.

5d fix a. o. typo.

Fixed in the revised figure (now **Figure 4**)

Alison June Beckett

Reviewer #3 (Remarks to the Author):

The authors present an impressive and innovative approach to multi-scale and multi-modal imaging. The relevance and interpretation for neuroscience is convincing.

The fact that the authors can bridge live imaging of the olfactory bulb to high resolution and volume EM, and show the advantage of XRT large volume overviews in choosing the regions of interest is very convincing and can pave the way for important future work. Image data is evaluated with high-end quantitative analysis and valuable details and protocols are described and disclosed, but some important information is missing, and not all of the quantifications are adequately described /presented (see below). In view of the tremendous amount of data and work compiled, it is not surprising that the organisation of material and figures presents some challenges.

Thank you for your detailed assessment, support and constructive comments. We have taken all comments on board, rearranged and streamlined the paper overall and importantly included substantially more detail on the SXRT including more analysis and more new data. We believe that this has substantially improved our manuscript and we are indeed grateful for the detailed and insightful comments.

I would like to recommend the work for publication, but can do so only subject to a major revision targeting two major points /deficits:

1.) Completeness of XRT description

Given the fact that the central claim of novelty is SXRT, and the correlative workflow, the SXRT is insufficiently described/quantified.

Thank you for pointing this out. We have now added all the points suggested and significantly expanded and quantified the SXRT imaging as detailed below.

Essential information such as the resolution achieved is missing:

-Fourier Shell Correlation and power spectral density should be used to estimate resolution for LXRT and SXRT (both beamlines); resolution is visibly much, much worse than the pixel, putting in question the choice of the objectives used

Thank you for this suggestion. We have now performed FSC for both the SBEM data and the SXRT. As shown in **Figure R_3.1** this indeed results in a resolution estimate of ~1.5-2.5 μm for the SXRT for both beamlines (Specific locations for FSC measurements are depicted in **appendix figures A1-A4**). We have also performed power spectral density analysis (**Figure R_3.2**). Our choice of objective (Olympus 10x UPlan S Apo at I13 and APO 20x at TOMCAT) was dictated by the available objectives (and a compromise between acquisition time and final image resolution). We have now performed additional experiments with a lower magnification objective at ID19 (resulting in pixel size of 650 nm, **Figure R_3.6** below) as well as a higher magnification objective at TOMCAT (**Figure R_3.3**). The lower magnification objective did indeed result in only minor decreases in resolvable features. For the higher magnification objective, there were some further improvements; these were, however, minimal, suggesting that resolution was dominated by other factors, possibly reconstruction / filtering (see below). This is indeed an important part that we now discuss in the revised manuscript in detail in the revised methods section on **page 12**. We also include the FSC analysis as a new **Supplementary Figure 7**, the PSD analysis as part of the new **Supplementary Figure 13**, and the comparison of different objectives as **Supplementary Figure 14** in the revised manuscript.

Figure R_3.1. Resolution of LXRT, SXRT and SBEM datasets.

Resolution of the datasets obtained across all structural modalities. **(a-c)** Volume ROIs large enough so relevant features would be contained were sampled from the original datasets (500 voxels wide for all cases except for LXRT, where ROIs were 100 voxels wide). For each trace, two volume ROIs acquired independently and containing data of the same volume were compared through FSC analysis against each other. In SXRT **(a)** and LXRT datasets **(c)** half-tomogram reconstructions were generated from odd and even projection subsets. In the SBEM dataset (originally acquired at 50 nm voxel size) **(b)** odd and even slices laterally downsampled by $\frac{1}{2}$ generated equivalent ROIs at 100nm isotropic voxel size. **(d-f)** Fourier shell correlation curves observed in the different imaging modalities. Variability was assessed across specimens, reconstruction algorithms and histological regions. For LXRT **(f)**, the reconstruction algorithm of the half-tomograms was not fully editable and possibly contained some common filtering step, which rendered artefactual high correlations at even $1/(2 \cdot \text{pixel})$ frequency. The average FSC curve (bold solid line) was normalized (bold dashed line) assuming the correlations at frequencies $[1/(3 \cdot \text{pixel}) \ 1/(2 \cdot \text{pixel})]$ (orange segment) would be null. The same normalization was applied to all individual traces (thin dashed lines) from which resolution measurements were extracted later on (green crosses). **(g-i)** Resolution readouts were obtained for each FSC curve at the point it crossed the resolution criteria for the different imaging modalities. Four commonly used resolution criteria were applied to most cases: 1 bit, $\frac{1}{2}$ bit, 3 sigma and 1/7.

Figure R_3.2 - Power spectral density

Power spectral densities were calculated from flat-field corrected phase maps for data obtained at TOMCAT (**a**, sample C432) and Diamond (**b**, sample C435). Top row (**a1-b1**): Full PSDs. Bottom row (**a2-b2**): Azimuthally-averaged PSDs.

Figure R_3.3 - SXRT at 20x and 40x

(a, b) SXRT datasets obtained with 20x (a) and 40x (b) objectives on the same specimen at the TOMCAT beamline. Full tile field of view is shown for each case. Both datasets were acquired with a distance of 50mm between sample and detector, and the tomograms reconstructed using Paganin filtering.

(c, d) Enlarged detail of the zones indicated in the 20x dataset (c) and 40x (d), respectively. Cell nuclei (asterisks), nucleoli (arrowheads) and dendritic processes (arrows) are clearly defined in both, with the 40x dataset enabling improved delineation of these features.

-For the heavy metalized samples phase contrast effects are not obvious, and should be illustrated by showing projections, and a reconstructed projection,

Thank you for pointing out this important issue. We have now explicitly calculated absorption for our samples (see above - **Figure R_1.3**). To test the potential contribution of phase contrast, at I13 we obtained projections at different distances (see above - **Figure R_1.4**). For larger distances indeed more information can be recovered (see e.g. the PSDs in **Figure R_1.4g**) indicative of a potential phase contrast contribution. Reconstructions at different distances (**Figure R_1.4a**) show that small distances (~absorption regime) do not allow us to resolve details to the same extent as longer propagation distances where the partial coherence of the beam allows for the use of phase contrast reconstruction (**Figure R_1.4b,c**). Thus, despite the (relatively) strong absorption, there is phase contrast information present in the data and e.g. the Paganin method allows us to recover improved images compared to the pure absorption regime.

To obtain a comparison with a beamline heavily relying on phase contrast effects we have now also performed experiments in the nano-holotomography beamline ID16A at ESRF (with the same type of sample, i.e. heavy metal stained, embedded olfactory bulb tissue, **Figure R_3.6**). This also allows for different phase retrieval approaches (see below).

Nevertheless, the phase information present in the data from the microtomography beamlines TOMCAT and I13 (both operating with only partially coherent beams) was beneficial for reconstruction (**Figure R_1.4**).

We now discuss this important point and explicitly show the different aspects of phase contrast imaging in the new **Supplementary Figures 12,13,15** and on **pages 13-14** of the revised manuscript.

-The assumption of a homogeneous object is probably not given. Please comment.

This is again an important point and we apologize that we did not discuss this properly in the previous version. We have now analysed the density variations in our samples, using “ground truth” electron microscopy data from a SBEM image of stained brain tissue. As can be seen in **Figure R_3.4**, density indeed varies on the sub-micrometer scale. However, firstly, density variations within tissue are in fact only about 22% ($SD_{\text{tissue}} / (\text{mean}_{\text{tissue}} - \text{mean}_{\text{void}})$). Secondly, when assessed at lower resolution ($(0.1\mu\text{m})^3$ - $(1\mu\text{m})^3$), heterogeneity decreases to 12% and 4%, respectively.

Finally, it has been argued that the Paganin reconstruction algorithm can be rather insensitive to inhomogeneities in the sample, provided that delta/beta ratios remain rather constant (e.g. ¹ - on their page 619: “If the object under study consists of a material that is not homogeneous but in which the fraction beta/delta is constant, the algorithm [Paganin] remains valid. Particular cases in which this relaxed condition is met are (i) objects with homogeneous elemental composition but varying density”). This condition is indeed met: We are staining our samples with exogenous heavy metals, the dominant sources of electron density are the same metals throughout the tissue in approximately unchanging ratios - albeit with different density. I.e. we expect lead, osmium and uranium to be present throughout the tissue but enriched (with higher density) at membranes. This robustness is also consistent with our experience that Paganin reconstruction can be robustly performed even for only partially coherent beams (see above).

We now discuss this important point on **pages 13/14** of the revised manuscript and include a new **Supp Fig 16** describing the density variations.

Figure R_3.4. Sample homogeneity.

(a) Backscattered electron image obtained with SBEM of the region at the edge of the blockface (stained brain tissue, mouse olfactory bulb, EPL). The field of view contains the void before the specimen begins, and the specimen composed of an external layer of resin with conductive silver followed by the stained tissue. The originally acquired image (top) was digitally downsampled 10 and 100-fold (bottom). The grey values of the horizontal midline of each image were recorded. (b) Signal recorded at the original and downsampled images.

- The assumption of direct contrast regime ($F > 1$) is not given for the chosen pixel size. Please give Fresnel numbers. Did you try different phase retrieval approaches, such as CTF valid also for lower F ?

Thank you for suggesting this. We have now calculated Fresnel numbers for the different beamlines and configurations used (Fig. R_3.5). We include this table as a new table in the revised manuscript to collect key relevant imaging parameters together in an efficient manner (Supplementary Table 2 in the revised manuscript).

	Diamond-I3-2	SLS-TOMCAT	ESRF-ID19	ESRF-ID16A
Scintillator	GGG:Eu 34 μm	LuAG:Ce 20 μm	LuAG:Ce 25 μm	GGG:Eu 23 μm
Camera	pco.edge 5.5	pco.edge 5.5	pco.edge 5.5	FreLoN 16
Exposure time (s)	0.4	0.3	0.1	0.33
Number of projections	3001	2601	1800	1900
Total rotation ($^{\circ}$)	180	180	180	180
Scan duration (min)	20	13	3	45
Field of view (μm)	832	832	1300	192
Effective pixel size (nm)	325	325	650	94
Effective distance (mm)	52	50	105	38.45
Energy (keV)	22	21	26	33.6
Wavelength (\AA)	0.56	0.59	0.477	0.369
Fresnel number	0.0363	0.0358	0.0844	0.0062

Figure R_3.5 - Experimental parameters at different beamlines.

Indeed, as the reviewer rightly points out, calculated Fresnel numbers are <1 , indicating that different phase retrieval methods such as CTF might be more appropriate. We had originally chosen Paganin reconstructions as these are known to be robust even with only partially coherent / monochromatic conditions. To directly assess the most suitable phase retrieval approach, we thus compared CTF and Paganin side-by-side. As can be seen from **Figure R_3.6**, while CTF has better representation of higher frequencies, reconstruction artefacts dominate the reconstruction of the only partially coherent I13 data. To compare CTF with Paganin in a situation with improved beam coherence we performed new experiments with identical samples at ID16A (highly coherent with 1% $\Delta E/E$, cf Figure R_3.11 for the full spectrum at I13-2). Indeed, under these circumstances (with $F \ll 1$), CTF markedly improves reconstruction (**Figure R_3.6**). However, many of the widely accessible μCT beamlines tend to prioritise high flux over high coherence (thus employing no or relatively broad bandwidth monochromators). In these cases the higher flux allows for rapid tomography, however at reduced temporal coherence. Under these circumstances we believe that Paganin allows for a more robust reconstruction. We now highlight these important points on **page 9** and in detail in the methods on **pages 13-14** of the revised manuscript and illustrate the differences with new **Supplementary Figures 12,13** and in the aforementioned summary **Supplementary Table 2**.

Figure R_3.6. Comparison of reconstructions with CTF and Paganin phase-retrieval. (a1) Reconstructed image, (a2) PSD, and (a3) azimuthally-averaged PSDs after paganin phase-retrieval for data from the microtomography beamline Diamond I13-2. (a4-a6) same as (a1-a3) but for CFT phase-retrieval. (b) same as (a) but for nano-holotomography at ESRF ID16-A (c) same as (a) but for the microtomography beamline ESRF ID19 for a 800 μm cylindrical sample.

-estimate delta and beta for different labeled components based on what is known for uranyl and osmium stoichiometry and concentration

Thank you for raising yet another very relevant point. Unfortunately, it is not known what the exact chemical composition is for soft tissue specimens after having undergone a staining protocol for electron microscopy. These protocols typically involve a sequence of staining steps separated by washes in aqueous solution aimed to enable the sequential binding of heavy metals (Os, U, Pb, Fe) to the proteins and lipids of the tissue. The staining process within each step can be complex - e.g. sequential oxidation-reduction processes in the microscale environment might allow Os to diffuse through the cell membranes made of phospholipid bilayers². Furthermore, some steps are meant to amplify the signal - e.g. thiocarbonylhydrazide, which upon binding to tissue-bound osmium it will offer several osmium-

binding sites, which can be used to bind more osmium³. These metals are usually dissolved in highly acidic solutions (pH ~3-4), which are likely to degrade some of the original fatty acids and proteins during the hour-long incubations at 4-50°C. Moreover, the load of heavy metals absorbed per unit of volume will depend on the tissue microanatomy, with noticeable changes within the sub-mm scale: brain regions packed with thin (<0.5 µm) axons, such as the olfactory nerve layer (onl), will have more membranes, therefore more fat, and hence incorporate more heavy metals than similar volumes of regions rich in wide (~5 µm) dendrites, such as the external plexiform layer, located adjacent to the onl (this feature will generate the contrast needed for exploiting the benefits of SXRT in tissue microanatomy studies). Finally, these protocols were aimed to confer plastic mechanical properties to the specimen, so it can be cut with ultramicrotomy techniques. This requirement imposes a last sequence of dehydration, resin-embedding and heat-accelerated resin curation steps that ultimately deliver a sample whose chemical composition differs significantly from the one of the original tissue and is challenging to predict.

We tried to measure the concentration of different chemicals in the stained tissue block. Unfortunately, using direct measurement approaches such as mass spectrometry was precluded by the small tissue volumes and the hardness and chemical inertness of the stained tissue and embedding materials.

In an effort to establish an initial hypothesis on what could be the simplest chemical composition of our stained samples, we have assumed that each sequential staining step would stain the sample until it reached a molar equilibrium with the new staining solution (**Figure R_3.7, R_3.8**).

staining step	component name	formula	mol. weight	conc.	units	conc. (mol/l)	diluent
Reduced Osmium	osmium tetroxide	OsO ₄	254.2	2%	w/v	0.079	sodium buffer cacodylate
Reduced Osmium	potassium hexacyanoferrate (II) / potassium ferrocyanide	K ₄ Fe(CN) ₆ · 3H ₂ O	422.39	3%	w/v	0.071	sodium buffer cacodylate
Reduced Osmium	calcium chloride	CaCl ₂	110.98	2	mM	0.002	sodium buffer cacodylate
Osmium	osmium tetroxide	OsO ₄	254.2	2%	w/v	0.079	H ₂ O
Thiocarbohydrazide	thiocarbohydrazide	(NH ₂ NH) ₂ CS	106.15	1%	w/v	0.094	H ₂ O
Uranyl acetate	uranyl acetate	C ₄ H ₆ O ₆ U	424.146	2%	w/v	0.047	H ₂ O
Lead aspartate	lead nitrate	Pb(NO ₃) ₂	331.2	0.02	M	0.020	H ₂ O
Lead aspartate	aspartic acid	C ₄ H ₇ NO ₄	133.1	0.03	M	0.030	H ₂ O
Resin	Epon / Epon 812	C ₂₁ H ₂₅ ClO ₅	329.9	48%	w/v	1.455	resin
Resin	DDSA / Dodeceny Succinic Anhydride Specially Distilled	C ₁₆ H ₂₆ O ₃	266.32	18%	w/v	0.676	resin
Resin	MNA / Methyl-5-Norbornene-2,3-Dicarboxylic Anhydride	C ₁₀ H ₁₀ O ₃	178.18	30%	w/v	1.684	resin
Resin	BDMA / Benzyl dimethylamine	C ₉ H ₁₃ N	135.21	2.8%	w/v	0.207	resin

Figure R_3.7. Chemical compounds used in the staining, dehydration and embedding protocol.

To that end, for each successive step, the mols of every element bound to the sample were calculated from the product of the molar concentration of the staining agent and the stoichiometry of that element in the agent. For example, the first staining step consisted in buffered 2% osmium tetroxide (OsO₄, 0.079M) reduced in 3% potassium ferrocyanide (K₄Fe(CN)₆ · 3H₂O, 0.071M) and 0.002M calcium chloride (CaCl₂). This step would therefore add 1*0.079 = 0.079 moles of Os, and 4*0.079 = 0.316 moles of O. The same reasoning allows calculating the moles of all the elements incorporated into the sample in this step (Os, O, K, Fe, C, N, Ca, Cl). Next, we added the moles of each element incorporated through all staining steps. For example, since 2% OsO₄ is used in two separate steps, each step would contribute in 0.079 moles of Os, and therefore the total addition of Os in the sample would be 2*0.079 = 0.158 moles. The same reasoning allows calculating the moles of all the elements incorporated into the sample throughout the entire staining protocol (H, C, N, O, S, Cl, K, Ca, Fe, Os, Pb, U).

Symbol	Stoichiometry	Atomic Mass	MW per element
C	60.902	12.011	731.494
O	15.509	15.999	248.129
H	74.538	1.007	75.060
Cl	1.459	35.453	51.726
Os	0.158	190.23	30.056
N	1.079	14.007	15.114
U	0.047	238.029	11.187
K	0.284	39.098	11.104
Pb	0.020	207.2	4.144
Fe	0.071	55.845	3.965
S	0.094	32.065	3.014
Ca	0.002	40.078	0.080

Figure R_3.8. Estimated average composition of the specimen.

The assumption made is that the contribution of each staining step is exclusively defined by the concentration of the staining solution.

Therefore, if the composition of the sample was only determined by addition of metals according to their concentration of each staining solution, the sample's formula would be (Figure R_3.8):

From here, one can calculate the molecular weight (MW) of the specimen

$$MW = \sum_{i=1}^n s_i * m_i = 1.1851 * 10^3 \text{ g/mol}$$

Where s_i indicates the stoichiometry and m_i the atomic mass of each element i with which the sample is stained.

With these details we could then estimate the X-ray transmission for different samples (Fig. R_3.9, sample asymmetry allowed us to measure transmission for different sample thicknesses).

Fig. R_3.9. Measured and estimated transmission.

From **Fig. R_3.9** it becomes clear that the samples absorbed more X-rays than expected from our calculations. This suggests that the concentration of heavy metals in the sample exceeded the one in the staining solutions. In fact, this has recently been pointed out by other studies, showing how stained brain tissue samples imaged with X-rays while in the staining solution appear darker than the solution⁴. Assuming all heavy staining agents (Os, Pb, U, Fe) are similarly sequestered into the tissue, we have attempted to estimate the degree of uptake by fitting the experimentally measured transmission curves with a scaling factor m over the stoichiometric concentrations. The best fit was achieved for $m=4.52\pm 1.04$ which we have used for subsequent calculations. To obtain experimental validation of this scaling feature we attempt in future work to use complementary X-ray techniques such as X-ray fluorescence to quantify local metal concentration. We believe, however, that this is outside the scope of this manuscript, not the least as quantitative density or elemental compositions are not essential to enable our analysis. The qualitative density maps we have achieved are wholly sufficient to segment the neuronal connections. Detailed elemental analysis, however, might help to further optimize staining procedures in the future and we discuss this important point on **page 9** of the revised manuscript.

To provide more insight into the sample properties, we have now determined density as well. To this end, we polymerised new batches of resin. Assuming that the density of resin-embedded brain tissue is similar to the density of the embedding material alone, the density can be estimated to be 1.24 ± 0.01 g/ml (**Fig. R_3.10**).

Figure R_3.10 - Density of hard epox resin.

Density was calculated from 3 independent batches of resin, with 5 resin blocks of similar volume each.

We then calculated the refractive index decrements β and δ using the chemical formula and density given above. The mean values for β and δ have been calculated from the weighted energy spectrum at I13 with $E_{\text{mean}} = 22.1$ keV. Using the adjustment factor m as defined above we obtain $\delta = 5.7 \cdot 10^{-7}$; $\beta = 6.2 \cdot 10^{-9}$, and $\delta/\beta = 98$.

It is worth noticing that these values differ from the δ/β we determined empirically (see above). This can be due to a number of reasons: The δ/β ratio chosen in the reconstruction was optimized not predominantly for homogeneous phase contrast but to facilitate the segmentation of neuronal structures. For larger δ/β ratios, the Paganin filter kernel (essentially a lowpass filter) also blurs the image significantly. While grey scale values might indeed become consistent with images expected from truly quantitative phase contrast approaches like holotomography, the concomitant low-pass filtering makes it harder to segment structures that are small relative to the pixel size (e.g. a few pixel wide). Lower δ/β ratio in turn preserves high spatial frequency features, albeit partially sacrificing quantitative interpretability of the resultant grey values.

This is indeed a very important point and a possible avenue for future optimization of the reconstruction process, in particular for sources with increasing coherence. We now discuss this in detail on **page 9** of the revised manuscript.

-give measured transmission for the samples & attenuation coefficients

We have now measured transmission for 6 samples for their shortest and their longest axis (see **Figure R_1.3.** above). Assuming homogeneous material we can then estimate the attenuation coefficient to $\mu = 1230 \pm 120 \text{ m}^{-1}$ ($\mu = -\ln(I/I_0)/x$, mean \pm sem). This compares to X-ray attenuation for osmium at our energy of 22 keV of $\mu_{\text{Os}} \approx 2 \cdot 10^5 \text{ m}^{-1}$ (e.g. ⁵).

-include a simulation of which contrast you would expect as a function of E (for the modeled optical tissue) so that choice of photon energy can be rationalized

Thank you for pointing this out. To calculate the contrast as a function of energy we note that the phase shift in the projection approximation is given by

$$\Delta\phi(x, y, z = z_0) = -k \delta T_{\text{sample}}(x, y)$$

Hereby we make the simplifying assumption of a homogeneous sample material (see above). We are aware that the condition of the Fresnel number $F > 1$ is not met (see above - for the case of I13-2 data: $\lambda=0.56 \cdot 10^{-10}$ m, $z_0=0.052$ m, $a=0.65 \cdot 10^{-6}$ m - this corresponds to 2 pixels on the camera -, $M=1$ resulting in $F=0.1465$). We therefore cannot readily extract quantitative results from these measurements. However, considering that the primary interest is in qualitative contrast in order to segment the neuronal connections, we argue that the choice of experimental parameters (and the above approximation) is reasonable and justified.

Based on this approximation, we have plotted X-ray flux, transmission and phase contrast for different parameters in **Figure R_3.11**. Using the mean energy of our beam, we calculate X-ray transmission for different sample thicknesses. Considering the slightly chromatic spectrum, transmissions are between 50% and 5% for the most extreme cases of elongated samples.

As an example for the phase shift, we performed calculations for a sample thickness of 3 mm and an assumed density difference of 2% (with the same overall estimated chemical formula - see above). This shows a phase shift of just larger than π ($\varphi=3.6872$ rad) for the selected mean energy. Thus, the energies available with the beamline I13 were indeed well suited to provide significant contrast for our metal-stained mm-size samples.

While indeed these estimates are consistent with our experimental observations, we are at this stage hesitant to include them in significant detail in the manuscript as estimating sample composition is notoriously difficult as highlighted above. We now mention this in the revised manuscript on **page 12**.

Figure R_3.11 Simulation of X-ray contrast.

- give details of the tomographic reconstruction including post-processing (such as ring filter)
 We apologize for this omission. We now include a detailed description of all post-processing steps in the methods section of the revised manuscript (**pages 12-13**). In brief, all data was saved in hdf5 container files. At I13, the reconstruction pipeline Savu⁶ was used for reconstructing the SRXT datasets. The full processing pipeline steps were: a. Loading of HDF5 dataset; b. Correction for dark and flat-field images; c. Paganin filter; d. Ring removal; e. Automatic center-finding; f. Reconstruction; g. Saving images as tiff files. Dark images, flat-field images and projections were extracted from the HDF5 dataset (a.) and all projections p were corrected for detector dark current d and the flat-field f intensity variation (b.). The normalized projections p_{norm} were calculated using: $p_{norm} = \frac{p-d}{f-d}$. The normalized projections were filtered with a Paganin filter⁷ using the following settings: detector distance 52 mm, X-ray energy 22 keV and detector pixel size 330 nm. Afterwards, ring artefacts were detected using algorithm #4 and rings were removed using the algorithms #6, #5, and #3 described in ⁸. For algorithms #6 and #5, a signal-to-noise ratio (SNR) of 3.0 and a filter window size of 71 pixel was used. The window size used for algorithm #3 was 31 pixels.

The center of reconstruction was determined using the algorithm from ⁸ and the reconstruction was performed using the GPU implementation of the Astra toolbox⁹⁻¹¹. A filtered back-projection algorithm with a Ram-Lak filter and outer padding (padding factor of $\sqrt{2}$) was applied to the Paganin-filtered, normalized projections. The resulting data was saved as 32-bit floating tiff images.

2.) Quality of the Presentation

The lack of detail and precision for the SXRT/LXRT is contrasted by information overflow in other parts of the MS /SM.

Generally I find that the data and method presentation is too packed, and not organized well enough. Think about using more tables, less crowded composite graphics; larger Figures, where essential (e.g. SXRT/LXRT slices); all essential graphical displays should be larger. In case of space constraints, leave something less important out.

Thank you very much for your constructive reading of our manuscript. Re-reading the manuscript we have to agree that we hadn't presented the data in the best manner with figures and text suboptimally structured and too packed. We have now substantially cleaned up the main figures, reducing clutter, increasing the size of relevant images whilst eliminating any redundant information and moving supporting figure parts into the supplement. Please see e.g. **Figures 2, 3, 4 & 6** where these edits might be most apparent. While we have added supplementary material providing important detail regarding the SXRT as outlined above, we have otherwise aimed to reduce and streamline the supporting information as well (e.g. in the description of the spine distributions as outlined below). Finally, we have polished the text and either removed peripheral information or relocated it to dedicated methods and supplementary information sections.

Supp.Fig.9, for example: As such, a1 and a2 are just not informative. Take one curve, explain it well and in detail, including acronyms of the legend, and then show the differences between different cells in form of a table.

We agree and have followed the suggestions of the reviewer.

I find the entire description of the inter-spine distribution is quite incomprehensible.

Upon re-reading this aspect we have to agree. We want to provide the inter-spine distribution as a relevant example and application of the correlative multimodal approach. We have now briefly and concisely described the approach and main findings as they relate to the rest of the paper.

As a test whether the presentation/description is fit for publication: ask each other within the co-author team, whether everybody can understand each other's part well.

This is an excellent suggestion and we have indeed now done exactly that. Furthermore we have asked colleagues unfamiliar with the work from both the synchrotron, EM and neuroscience field for comments that have been very helpful. We are convinced that the revised manuscript is significantly better in presentation, style, and clarity and thank the reviewer for their constructive suggestions.

Fig.5, for example: (i) I cannot find the description, p-values refer to which test? what is the finding exactly? What do we learn, by plotting SA density as a function of some depth?

Any hypothesis or interpretation associated with this?

We have now clarified these points as suggested. Statistical tests are referred to in the respective methods section for each measurement, and presented measurements have been streamlined or recalculated more homogeneously to ensure readability (e.g. see **Supp. Fig. 9**). We have also extended the discussion of the implications of differential SA density in dendrites belonging to deep vs superficial CA1 neurons in **page 8**.

(g,j) The segmentations of the high res EM are not explained, what are we seeing. Of course, once you know how a spine apparatus looks in EM, you can guess; but overall please carefully choose, describe, and if in doubt leave out. The high res EM of the spines is nice, why not present it larger at the expense of something else.

We have moved less central aspects of the figures to supplement and highlighted the key aspects of this part of the study as described above. We have decluttered this figure (now **Figure 4**) and have therefore been able to magnify the high-resolution images accordingly.

Sometimes less is more.

Indeed - and we hope we have now embraced this in the revised manuscript.

Appendix - locations of FSC measurements:

Figure R_A1 - ROIs extracted from an individual tile of two specimens imaged at DIAMOND I13-2

(a) Specimen C525, (b) specimen C432. For each tile, the raw projections were split into two data series containing the odd and even projections, respectively, and the two half-tomograms were reconstructed independently. Therefore, each half-tomogram contained independently-acquired information of the same volume. Three 500 voxel-wide volume ROIs were extracted from each reconstruction and paired with their related ROIs of the other half-tomogram for further FSC analysis. ROIs locations in the tile were checked to ensure they contained tissue data across all slices (right panels).

a

b

c

100 μ m

100 μ m

Figure R_A2 - ROIs extracted from an individual tile of an specimen (C432) imaged at TOMCAT.

The raw projections were split into two data series containing the odd and even projections, respectively, and the two half-tomograms were reconstructed independently. Therefore, each half-tomogram contained independently-acquired information of the same volume. Reconstructions were performed following three algorithms: using Paganin filtering (a), using Paganin and an additional ring correction step (b), and not using Paganin filtering (c). Three 500 voxel-wide volume ROIs were extracted from each reconstruction and paired with their related ROI of the other half-tomogram for further FSC analysis. ROI locations in the tile were checked to ensure they contained tissue data across all slices (right panels).

Figure R_A3. ROIs extracted from the registered C525b dataset imaged with SBEM.

Three histological zones were sampled: external plexiform layer (a), glomerular layer (b) and olfactory nerve layer (c). Three volume data samples were extracted from each zone. Each sample was split into two data series containing the odd and even slices, respectively, and downsampled by $\frac{1}{2}$ in x,y. This provided two 500 voxel-wide volume rois with an isotropic voxel size of 100nm, each containing independently-acquired information of the same volume. Roi locations were checked to ensure they contained tissue data across all slices.

Figure R_A4. ROIs extracted from a specimen (C435) imaged with LXRT.

The raw projections were split into two data series containing the odd and even projections, respectively, and the two half-tomograms were reconstructed independently using proprietary software. Therefore, each half-tomogram contained independently-acquired information of the same volume. Three 100 voxel-wide volume ROIs were extracted from each reconstruction and paired with their related roi of the other half-tomogram for further FSC analysis. ROI locations in the tile were checked to ensure they contained tissue data across all slices (right panels).

References

- 1 Weitkamp, T., Haas, D., Wegrzynek, D. & Rack, A. ANKAphase: software for single-distance phase retrieval from inline X-ray phase-contrast radiographs. *J Synchrotron Radiat* **18**, 617-629, doi:10.1107/S0909049511002895 (2011).
- 2 Hua, Y., Laserstein, P. & Helmstaedter, M. Large-volume en-bloc staining for electron microscopy-based connectomics. *Nature communications* **6**, 7923, doi:10.1038/ncomms8923 (2015).
- 3 Mikula, S. & Denk, W. High-resolution whole-brain staining for electron microscopic circuit reconstruction. *Nature methods* **12**, 541-546, doi:10.1038/nmeth.3361 (2015).
- 4 Ströh, S., Hammerschmith, E. W., Tank, D. W., Seung, H. S. & Wanner, A. A. In situ X-ray assisted electron microscopy staining for large biological samples. *bioRxiv*, 2021.2006.2019.448808, doi:10.1101/2021.06.19.448808 (2021).
- 5 Müller, B. *et al.* in *Proc.SPIE*.
- 6 Wadson, N. & Basham, M. Savu: a Python-based, MPI framework for simultaneous processing of multiple, N-dimensional, large tomography datasets. *arXiv*, doi:arXiv:1610.08015 (2016).
- 7 Paganin, D., Mayo, S. C., Gureyev, T. E., Miller, P. R. & Wilkins, S. W. Simultaneous phase and amplitude extraction from a single defocused image of a homogeneous object. *J Microsc* **206**, 33-40 (2002).
- 8 Vo, N. T., Atwood, R. C. & Drakopoulos, M. Superior techniques for eliminating ring artifacts in X-ray micro-tomography. *Opt Express* **26**, 28396-28412, doi:10.1364/OE.26.028396 (2018).
- 9 van Aarle, W. *et al.* The ASTRA Toolbox: A platform for advanced algorithm development in electron tomography. *Ultramicroscopy* **157**, 35-47, doi:10.1016/j.ultramic.2015.05.002 (2015).
- 10 van Aarle, W. *et al.* Fast and flexible X-ray tomography using the ASTRA toolbox. *Opt Express* **24**, 25129-25147, doi:10.1364/OE.24.025129 (2016).
- 11 Palenstijn, W. J., Batenburg, K. J. & Sijbers, J. Performance improvements for iterative electron tomography reconstruction using graphics processing units (GPUs). *J Struct Biol* **176**, 250-253, doi:10.1016/j.jsb.2011.07.017 (2011).

REVIEWERS' COMMENTS

Reviewer #1 (Remarks to the Author):

Dear Editor,

I am completely satisfied with the revised manuscript. The authors replied to all my concerns providing additional data and explanations that enrich and improve the final paper.

In my view the revised manuscript is now suitable for publication in Nature Communications.

Reviewer #3 (Remarks to the Author):

I congratulate the authors for this careful and convincing reply and revision.